# Algorithm Selection for Deep Active Learning with Imbalanced Datasets

**Jifan Zhang**
University of Wisconsin - Madison
Madison, WI 53715
jifan@cs.wisc.edu

**Shuai Shao**
Meta Inc.
Menlo Park, CA 94025
sshao@meta.com

**Saurabh Verma**
Meta Inc.
Menlo Park, CA 94025
saurabh08@meta.com

**Robert Nowak**
University of Wisconsin - Madison
Madison, WI 53715
rdnowak@wisc.edu

## Abstract

Label efficiency has become an increasingly important objective in deep learning applications. Active learning aims to reduce the number of labeled examples needed to train deep networks, but the empirical performance of active learning algorithms can vary dramatically across datasets and applications. It is difficult to know in advance which active learning strategy will perform well or best in a given application. To address this, we propose the first adaptive algorithm selection strategy for deep active learning. For any unlabeled dataset, our (meta) algorithm `TAILOR` (**T**hompson **ActI**ve **L**earning alg**OR**ithm selection) iteratively and adaptively chooses among a set of candidate active learning algorithms. `TAILOR` uses novel reward functions aimed at gathering class-balanced examples. Extensive experiments in multi-class and multi-label applications demonstrate `TAILOR` 's effectiveness in achieving accuracy comparable or better than that of the best of the candidate algorithms. Our implementation of TAILOR is open-sourced at `https://github.com/jifanz/TAILOR`.

## 1 Introduction

Active learning (AL) aims to reduce data labeling cost by iteratively and adaptively finding informative unlabeled examples for annotation. Label-efficiency is increasingly crucial as deep learning models require large amount of labeled training data. In recent years, numerous new algorithms have been proposed for deep active learning [Sener and Savarese, 2017, Gal et al., 2017, Ash et al., 2019, Kothawade et al., 2021, Citovsky et al., 2021, Zhang et al., 2022]. Relative label efficiencies among algorithms, however, vary significantly across datasets and applications [Beck et al., 2021, Zhan et al., 2022]. When it comes to choosing the best algorithm for a novel dataset or application, practitioners have mostly been relying on educated guesses and subjective preferences. Prior work [Baram et al., 2004, Hsu and Lin, 2015, Pang et al., 2018] have studied the online choice of active learning algorithms for linear models, but these methods become ineffective in deep learning settings (see Section 2). In this paper, we present the first principled approach for automatically selecting effective *deep* AL algorithms for novel, unlabeled datasets.

We reduce the algorithm selection task to a multi-armed bandit problem. As shown in Figure 1, the idea may be viewed as a meta algorithm adaptively choosing among a set of candidate AL algorithms (arms). The objective of the meta algorithm is to maximize the cumulative reward incurred from running the chosen candidate algorithms. In Section 4, we propose reward functions that encourage

37th Conference on Neural Information Processing Systems (NeurIPS 2023).

the collection of *class-balanced* labeled set. As mentioned above, deep AL algorithms are generally proposed to maximize different notions of informativeness. As a result, by utilizing our algorithm selection strategy `TAILOR` , we annotate examples that are both *informative* and *class-diverse*.

To highlight some of our results, as shown in Figure 2 for the CelebA dataset, `TAILOR` outperforms all candidate deep AL algorithms and collects the least amount of labels while reaching the same accuracy level. `TAILOR` achieves this by running a combination of candidate algorithms (see Appendix E) to yield an informative and class-diverse set of labeled examples (see Figure 3(c)).

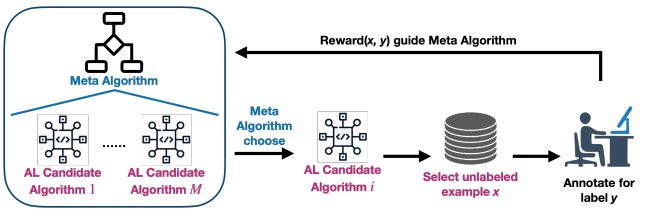

Figure 1: Adaptive active learning algorithm selection framework for batch size 1. Our framework proposed in Section 3.2 is a batched version that chooses multiple candidate algorithms and unlabeled examples in every iteration. Labels and rewards are revealed all at once at the end of the iteration.

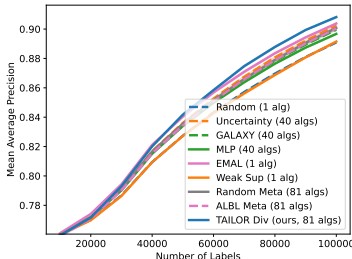

Figure 2: Mean Average Precision, CelebA

Our key contributions are as follows:

- To our knowledge, we propose the first adaptive algorithm selection strategy for *deep* active learning. Our algorithm `TAILOR` works particularly well on the challenging and prevalent class-imbalance settings [Kothawade et al., 2021, Emam et al., 2021, Zhang et al., 2022].
- Our framework is general purpose for both multi-label and multi-class classification. Active learning is especially helpful for multi-label classification due to the high annotation cost of obtaining *multiple* labels for each example.
- `TAILOR` can choose among large number (e.g. hundreds) of candidate deep AL algorithms even under limited (10 or 20) rounds of interaction. This is particularly important since limiting the number of model retraining steps and training batches is essential in large-scale deep active learning [Citovsky et al., 2021].
- In Section 5, we provide regret analysis of `TAILOR` . Although `TAILOR` can be viewed as a sort of contextual bandit problem, our regret bound is better than that obtained by a naive reduction to a linear contextual bandit reduction [Russo and Van Roy, 2014].
- We provide extensive experiments on four multi-label and six multi-class image classification datasets (Section 6). Our results show that `TAILOR` obtains accuracies comparable or better than the best candidate strategy for nine out of the ten datasets. On all of the ten datasets, `TAILOR` succeeds in collecting datasets as class-balanced as the best candidate algorithm. Moreover, with a slightly different reward function designed for active search, `TAILOR` performs the best in finding the highest number of positive class labels on all multi-label datasets.

## 2   Related Work

**Adaptive Algorithm Selection in Active Learning.** Several past works have studied the adaptive selection of active learning algorithms for linear models. Donmez et al. [2007] studied the limited setting of switching between two specific strategies to balance between uncertainty and diversity. To choose among off-the-shelf AL algorithms, Baram et al. [2004] first proposed a framework that reduced the AL algorithm selectino task to a multi-armed bandit problem. That approach aims to maximize the cumulative reward in a Classification Entropy Maximization score, which measures the class-balancedness of predictions on unlabeled examples, after training on each newly labeled example. However, this becomes computationally intractable for large datasets with computationally expensive models. To remedy this problem, Hsu and Lin [2015] and Pang et al. [2018] proposed the use of importance weighted training accuracy scores for each newly labeled example. The training accuracy, however, is almost always $100\%$ for deep learning models due to their universal approximation capability, which makes the reward signals less effective. Moreover, Hsu and Lin

[2015] reduced their problem to an adversarial multi-armed bandit problem while Pang et al. [2018] also studied the non-stationarity of rewards over time.

Lastly, we would like to distinguish the goal of our paper from the line of *Learning Active Learning* literature [Konyushkova et al., 2017, Shao et al., 2019, Zhang et al., 2020, Gonsior et al., 2021, Löffler and Mutschler, 2022], where they learn a single paramtereized policy model from offline datasets. These policies can nonetheless serve as individual candidate algorithms, while TAILOR aims to select the best subsets that are adapted for novel dataset instances.

**Multi-label Deep Active Learning.** Many active learning algorithms for multi-label classification based on linear models have been proposed [Wu et al., 2020], but few for deep learning. Multi-label active learning algorithms are proposed for two types of annotation, *example-based* where all associated labels for an example are annotated, and *example-label-based* where annotator assigns a binary label indicating whether the example is positive for the specific label class.

While Citovsky et al. [2021], Min et al. [2022] both propose deep active learning algorithms for example-label-based labels, we focus on example-based annotation in this paper. To this end, Ranganathan et al. [2018] propose an uncertainty sampling algorithm that chooses examples with the lowest class-average cross entropy losses after trained with weak supervision. We find the EMAL algorithm by Wu et al. [2014] effective on several datasets, despite being proposed for linear models. EMAL is based on simple uncertainty metric where one averages over binary margin scores for each class. Lastly, a multi-label task can be seen as individual single-label binary classification tasks for each class [Boutell et al., 2004]. By adopting this view, one can randomly interleave the above-mentioned AL algorithms for every class. In this paper, we include baselines derived from least confidence sampling [Settles, 2009], GALAXY [Zhang et al., 2022] and most likely positive sampling [Warmuth et al., 2001, 2003, Jiang et al., 2018].

**Balanced Multi-class Deep Active Learning.** Traditional uncertainty sampling algorithms have been adopted for deep active learning. These algorithms select uncertain examples based on scores derived from likelihood softmax scores, such as margin, least confidence and entropy [Tong and Koller, 2001, Settles, 2009, Balcan et al., 2006, Kremer et al., 2014]. The latter approaches leverage properties specific to neural networks by measuring uncertainty through dropout [Gal et al., 2017], adversarial examples [Ducoffe and Precioso, 2018] and neural network ensembles [Beluch et al., 2018]. Diversity sampling algorithms label examples that are most different from each other, based on similarity metrics such as distances in penultimate layer representations [Sener and Savarese, 2017, Geifman and El-Yaniv, 2017, Citovsky et al., 2021] or discriminator networks [Gissin and Shalev-Shwartz, 2019]. Lastly, gradient embeddings, which encode both softmax likelihood and penultimate layer representation, have become widely adopted in recent approaches [Ash et al., 2019, 2021, Wang et al., 2021, Elenter et al., 2022, Mohamadi et al., 2022]. As an example, Ash et al. [2019] uses k-means++ to query a diverse set of examples in the gradient embedding space.

**Unbalanced Multi-class Deep Active Learning.** More general and prevalent scenarios, such as unbalanced deep active classification, have received increasing attention in recent years [Kothawade et al., 2021, Emam et al., 2021, Zhang et al., 2022, Coleman et al., 2022, Jin et al., 2022, Aggarwal et al., 2020, Cai, 2022]. For instance, Kothawade et al. [2021] label examples with gradient embeddings that are most similar to previously collected rare examples while most dissimilar to out-of-distribution ones. Zhang et al. [2022] create linear one-vs-rest graphs based on margin scores. To collect a more class-diverse labeled set, GALAXY discovers and labels around the optimal uncertainty thresholds through a bisection procedure on shortest shortest paths.

## 3 Problem Statement

### 3.1 Notation

In pool based active learning, one starts with a large pool of $N$ unlabeled examples $X = \{x_1, x_2, ..., x_N\}$ with corresponding ground truth labels $Y = \{y_1, y_2, ..., y_N\}$ initially unknown to the learner. Let $K$ denote the total number of classes. In multi-label classification, each label $y_i$ is denoted as $y_i \in \{0, 1\}^K$ with each element $y_{i,j}$ representing the binary association between class $j$ and example $x_i$. On the other hand, in a multi-class problem, each label $y_i \in \{e_j\}_{j \in [K]}$ is denoted by a canonical one-hot vector, where $e_j$ is the $j$-th canonical vector representing the $j$-th class. Furthermore, at any time, we denote labeled and unlabeled examples by $L, U \subset X$ correspondingly,

where $L \cap U = \emptyset$. We let $L_0 \subset X$ denote a small seed set of labeled examples and $U_0 = X \backslash L_0$ denote the initial unlabeled set. Lastly, an active learning algorithm $\mathcal{A}$ takes as input a pair of labeled and unlabeled sets $(L, U)$ and returns an unlabeled example $\mathcal{A}(L, U) \in U$.

## 3.2 Adaptive Algorithm Selection Framework

In this section, we describe a generic framework that encompasses the online algorithm selection settings in Baram et al. [2004], Hsu and Lin [2015] and Pang et al. [2018]. As shown in Algorithm 1, the meta algorithm has access to $M$ candidate algorithms. At the beginning of any round $t$, a multi-set of $B$ algorithms are chosen, where the same algorithm can be chosen multiple times. One example is selected by each algorithm in the multiset sequentially, resulting in a total of $B$ unique examples. The batch of examples are then labeled all at once. At the end of the round, their corresponding rewards are observed based on the newly annotated examples $\{(x^{t,j}, y^{t,j})\}_{j=1}^{B}$ selected by the algorithms. The model is also retrained on labeled examples before proceeding to the next round.

Overall, the meta algorithm aims to maximize the future cumulative reward based on noisy past reward observations of each candidate algorithm. Th reward function $r : X \times Y \to \mathbb{R}$ is measured based on an algorithm's selected examples and corresponding labels. There are two key components to this framework: the choice of reward function and a bandit strategy that optimizes future rewards. Our particular design will be presented in Section 4.

---

**Algorithm 1** General Meta Active Learning Framework for Baram et al. [2004], Hsu and Lin [2015], Pang et al. [2018]

---

**Define:** $M$ candidate algorithms $A = \{\mathcal{A}_i\}_{i \in [M]}$, pool $X$, total number of rounds $T$, batch size $B$.
**Initialize:** Labeled seed set $L_0 \subset X$, unlabeled set $U_0 = X \backslash L_0$ and initial policy $\Pi^1$.
**for** $t = 1, ..., T$ **do**
    Meta algorithm $\Pi^t$ chooses multiset of algorithms $\mathcal{A}_{\alpha_{t,1}}, \mathcal{A}_{\alpha_{t,2}}, ..., \mathcal{A}_{\alpha_{t,B}}$, where indexes $\alpha_{t,1}, ..., \alpha_{t,B} \in [M]$. Initialize selection set $S_t \leftarrow \emptyset$.
    **for** $j = 1, ..., B$ **do**
        Run algorithm to select unlabeled example $x^{t,j} := \mathcal{A}_{\alpha_{t,j}}(L_{t-1}, U_{t-1} \backslash S_t)$ that is unselected.
        Insert the example $x^{t,j}$: $S_t \leftarrow S_t \cup \{x^{t,j}\}$.
    **end for**
    Annotate $\{x^{t,j}\}_{j=1}^{B}$ and observe labels $\{y^{t,j}\}_{j=1}^{B}$. Update sets $L_t \leftarrow L_{t-1} \cup S_t$, $U_t \leftarrow U_{t-1} \backslash S_t$.
    Observe reward $r^{t,j} = r(x^{t,j}, y^{t,j})$ for each algorithm $\mathcal{A}_{\alpha_{t,j}}$, where $j \in [B]$.
    Update policy statistics based on $x^{t,j}, y^{t,j}$ and $r^{t,j}$ to obtain $\Pi^{t+1}$ and retrain model on $L_t$.
**end for**
**Objective:** Maximize cumulative reward $\sum_{t=1}^{T} \sum_{j=1}^{B} r^{t,j}$.

---

We make the following two crucial assumptions for our framework:

**Assumption 3.1.** Any candidate batch active learning algorithm $\bar{A}$ can be decomposed into an iterative selection procedure $A$ that returns one unlabeled example at a time.

The assumption has been inherently made by our framework above where an deep active learning algorithm returns one unlabeled example at a time. It entails that running $\bar{A}$ once to collect a batch of $B$ examples is equivalent with running the iterative algorithm $A$ for $B$ times. As noted in Appendix A.1, most existing deep active learning algorithms can be decomposed into iterative procedures and thus can serve as candidate algorithms in our framework.

**Assumption 3.2.** For each round $t \in [T]$, we assume there exist ground truth reward distributions $\mathbb{P}_{t,1}, ..., \mathbb{P}_{t,M}$ for each candidate algorithm. Furthermore, for each element $j \in [B]$ in the batch, we make the iid assumption that reward $r^{t,j} \overset{iid}{\sim} \mathbb{P}_{t,\alpha_{t,j}}$ is sampled from the distribution of the corresponding selected algorithm.

The iid assumption is made for theoretical simplicity by all of Baram et al. [2004], Hsu and Lin [2015], Pang et al. [2018]. We say the distributions are *non-stationary* if for any $i \in [M]$, $P_{t,i}$ varies with respect to time $t$. Both this paper and Pang et al. [2018] study *non-stationary* scenarios, whereas Baram et al. [2004] and Hsu and Lin [2015] assume the distributions are *stationary* across time.

# 4 Thompson Active Learning Algorithm Selection

In this section, we present the two key components of our design, reward function and bandit strategy. In Section 4.1, we first present a class of reward functions designed for deep active learning under class imbalance. In Section 4.2, by leveraging the structure of such reward functions, we reduce the adaptive algorithm selection framework from Section 3.2 into a novel bandit setting. In Section 4.3, we then propose our algorithm `TAILOR` which is specifically designed for this setting. When using `TAILOR` on top of deep AL algorithms, the annotated examples are *informative* and *class-diverse*.

## 4.1 Reward Function

We propose reward functions that encourage selecting examples so that every class is well represented in the labeled dataset, ideally equally represented or "class-balanced". Our reward function works well even under practical scenarios such as limited number of rounds and large batch sizes [Citovsky et al., 2021]. The rewards we propose can be efficiently computed example-wise as opposed to Baram et al. [2004] and are more informative and generalizable than Hsu and Lin [2015] and Pang et al. [2018]. Our class-balance-based rewards are especially effective for datasets with underlying class imbalance. Recall $y \in \{0, 1\}^K$ for multi-label classification and $y \in \{e_i\}_{i=1}^K$ for multi-class classification. We define the following types of reward functions.

- **Class Diversity**: To encourage better class diversity, we propose a reward that inversely weights each class by the number of examples already collected. For each round $t \in [T]$,

$$r_{div}^t(x, y) = \frac{1}{K} \sum_{i=1}^{K} \frac{1}{\max(1, \text{COUNT}^t(i))} y_{:i} =: \langle v_{div}^t, y \rangle$$

  where $\text{COUNT}^t(i)$ denotes the number of examples in class $i$ after $t - 1$ rounds and $y_{:i}$ denotes the $i$-th element of $y$. We let $v_{div}^t$ denote the inverse weighting vector.
- **Multi-label Search**: As shown in Table 1, multi-label classification datasets naturally tend to have sparse labels (more 0's than 1's in $y$). Therefore, it is often important to search for positive labels. To encourage this, we define a stationary reward function for multi-label classification:

$$r_{search}(x, y) = \frac{1}{K} \sum_{i=1}^{K} y_{:i} =: \langle v_{pos}, y \rangle \quad \text{where } v_{pos} = \frac{1}{K} \vec{1}.$$

- **Domain Specific**: Lastly, we would like to note that domain experts can define specialized weighting vectors of different classes $v_{dom}^t \in [-\frac{1}{K}, \frac{1}{K}]^K$ that are adjusted over time $t$. The reward function simply takes the form $r_{dom}^t(x, y) = \langle v_{dom}^t, y \rangle$. As an example of multi-label classification of car information, one may prioritize classes of car brands over classes of car types, thus weighting each class differently. They can also adjust the weights over time based on needs.

## 4.2 Novel Bandit Setting

We now present a novel bandit reduction that mirrors the adaptive algorithm selection framework under this novel class of rewards. In this setup, $v_t \in [-\frac{1}{K}, \frac{1}{K}]^K$ is arbitrarily chosen and non-stationary. On the other hand, for each candidate algorithm $\mathcal{A}_i \in A$, we assume the labels $y$ are sampled iid from a stationary 1-sub-Gaussian distribution $\mathbb{P}_{\theta^i}$ with mean $\theta^i$. Both the stationary assumption in $P_{\theta^i}$ and the iid assumption are made for simplicity of our theoretical analysis only. We will describe our implementation to overcome the non-stationarity in $\mathbb{P}_{\theta^i}$ in Section 6.1. Although we make the iid assumption analogous to Assumption 3.2, we demonstrate the effectiveness of our algorithm in Section 6 through extensive experiments. Additionally, note that $\theta^i \in [0, 1]^K$ for multi-label classification and $\theta^i \in \Delta^{(K-1)}$ takes value in the $K$ dimensional probability simplex for multi-class classification. In our bandit reduction, at each round $t$,

1. Nature reveals weighting vector $v^t$;
2. Meta algorithm chooses algorithms $\alpha^{t,1}, ..., \alpha^{t,B}$, which sequentially select unlabeled examples;
3. Observe batch of labels $y^{t,1}, ..., y^{t,B}$ all at once, where $y^{t,j} \overset{iid}{\sim} \mathbb{P}_{\theta^{\alpha_{t,j}}}$;
4. Objective: maximize rewards defined as $r^{t,j} = \langle v^t, y^{t,j} \rangle$.

This setting bears resemblance to a linear contextual bandit problem. Indeed, one can formulate such a problem close to our setting by constructing arms $\phi_i^t = \text{vec}(v^t e_i^\top) \in [-\frac{1}{K}, \frac{1}{K}]^{KM}$. Here, $\text{vec}(\cdot)$ vectorizes the outer product between $v^t$ and the $i$-th canonical vector $e_i$. A contextual bandit algorithm observes reward $r = \langle \phi_i^t, \theta^\star \rangle + \varepsilon$ after pulling arm $i$, where $\theta^\star = \text{vec}([\theta^1, \theta^2, ..., \theta^M]) \in [0, 1]^{KM}$ and $\varepsilon$ is some sub-Gaussian random noise. However, this contextual bandit formulation does not take into account the observations of $\{y^{t,j}\}_{j=1}^B$ at each round, which are direct realizations of $\theta^1, ..., \theta^M$. In fact, standard contextual bandit algorithms usually rely on least squares estimates of $\theta^1, ..., \theta^M$ based on the reward signals [Russo and Van Roy, 2014]. As will be shown in Proposition 5.1, a standard Bayesian regret upper bound from Russo and Van Roy [2014] is of order $\widetilde{O}(BM^{\frac{3}{4}}K^{\frac{3}{4}}\sqrt{T})$. Our algorithm TAILOR , on the other hand, leverages the observations of $y^{t,j} \sim \mathbb{P}_{\theta^{\alpha^{t,j}}}$ and has regret upper bounded by $\widetilde{O}(B\sqrt{MT})$ (Theorem 5.2), similar to a stochastic multi-armed bandit.

---

**Algorithm 2** TAILOR : Thompson Active Learning Algorithm Selection

---

**Input:** $M$ candidate algorithms $A = \{\mathcal{A}_i\}_{i \in [M]}$, pool $X$, total number of rounds $T$, batch size $B$.

**Initialize:** For each $i \in [M]$, $a^i = b^i = \vec{1} \in \mathbb{R}^{+K}$.

**for** $t = 1, ..., T$ **do**
    Nature reveals $v^t \in [-\frac{1}{K}, \frac{1}{K}]^K$.
    **Choose candidate algorithms:**
    **for** $j = 1, ..., B$ **do**
        For each $i \in [M]$, sample $\widehat{\theta^i} \sim \text{Beta}(a^i, b^i)$ for multi-label or $\widehat{\theta^i} \sim \text{Dir}(a^i)$ for multi-class.
        Choose $\alpha^{t,j} \leftarrow \arg\max_{i \in [M]} \langle v^t, \widehat{\theta^i} \rangle$.
    **end for**
    **Run chosen algorithms to collect batch:**
    **for** $j = 1, ..., B$ **do**
        Run algorithm $\mathcal{A}_{\alpha^{t,j}}$ to select unlabeled example $x^{t,j}$ and insert into $S_t$.
    **end for**
    Annotate examples in $S_t$ to observe $y^{t,j}$ for each $j \in [B]$.
    **Update posterior distributions:**
    For each algorithm $i \in [M]$:    $a^i \leftarrow a^i + \sum_{j:\alpha^{t,j}=i} y^{t,j}$,    $b^i \leftarrow b^i + \sum_{j:\alpha^{t,j}=i}(1 - y^{t,j})$.
    Retrain neural network to inform next round.
**end for**

---

### 4.3 TAILOR

We are now ready to present TAILOR , a Thompson Sampling [Thompson, 1933] style meta algorithm for adaptively selecting active learning algorithms. The key idea is to maintain posterior distributions for $\theta^1, ..., \theta^M$. As shown in Algorithm 2, at the beginning we utilize uniform priors $\text{Unif}(\Omega)$ over the support $\Omega$, where $\Omega = \Delta^{(t-1)}$ and $[0, 1]^K$ respectively for multi-label and multi-class classification. We note that the choice of uniform prior is made so that it is general purpose for any dataset. In practice, one may design more task-specific priors.

Over time, we keep an posterior distribution over each ground truth mean $\theta^i$ for each algorithm $i \in [M]$. With a uniform prior, the posterior distribution is an instance of either element-wise Beta distribution[1] for multi-label classification or Dirichlet distribution for multi-class classification. During each round $t$, we draw samples from the posteriors, which are then used to choose the best action (i.e., candidate algorithm) that has the largest predicted reward. After the batch of $B$ candidate algorithms are chosen, we then sequentially run each algorithm to collect the batch of unlabeled examples. Upon receiving the batch of annotations, we then update the posterior distribution for each algorithm. Lastly, the neural network model is retrained on all labeled examples thus far.

## 5 Analysis

In this section, we present regret upper bound of TAILOR and compare against a linear contextual bandit upper bound from Russo and Van Roy [2014]. Our time complexity analysis is in Appendix 5.1.

---

[1] For $z \in [0, 1]^d$ and $a, b \in \mathcal{Z}^{+d}$, we say $z \sim \text{Beta}(a, b)$ if for each $i \in [d]$, $z_i \sim \text{Beta}(a_i, b_i)$.

Given an algorithm $\pi$, the *expected regret* measures the difference between the expected cumulative reward of the optimal action and the algorithm's action. Formally for any fixed instance with $\Theta = \{\theta^1, ..., \theta^M\}$, the *expected regret* is defined as

$$R(\pi, \Theta) := \mathbb{E}\left[\sum_{t=1}^{T}\sum_{j=1}^{B}\max_{i\in[M]}\langle v^t, \theta^i - \theta^{\alpha^{t,j}}\rangle\right]$$

where the expectation is over the randomness of the algorithm, e.g. posterior sampling in `TAILOR` .

*Bayesian regret* simply measures the average of expected regret over different instances

$$BR(\pi) := \mathbb{E}_{\theta^i \sim \mathbb{P}_0(\Omega), i\in[M]}\left[R(\pi, \{\theta^i\}_{i=1}^{M})\right]$$

where $\Omega$ denotes the support of each $\theta^i$ and $\mathbb{P}_0(\Omega)$ denotes the prior. Recall $\Omega = [0,1]^K$ for multi-label classification and $\Omega = \Delta^{(K-1)}$ for multi-class classification. While `TAILOR` is proposed based on uniform priors $\mathbb{P}_0(\Omega) = \text{uniform}(\Omega)$, our analysis in this section holds for arbitrary $\mathbb{P}_0$ as long as the prior and posterior updates are modified accordingly in `TAILOR` .

First, we would like to mention a Bayesian regret upper bound for the contextual bandit formulation mentioned in 4.2. This provides one upper bound for `TAILOR` . As mentioned, the reduction to a contextual bandit is valid, but is only based on observing rewards and ignores the fact that `TAILOR` observes rewards and the full realizations $y^{t,j}$ of $\theta^{\alpha^{t,j}}$ that generate them. So one anticipates that this bound may be loose.

**Proposition 5.1** (Russo and Van Roy [2014]). *Let $\pi_{context}$ be the posterior sampling algorithm for linear contextual bandit presented in Russo and Van Roy [2014], the Bayesian regret is bounded by*

$$BR(\pi_{context}) \leq \widetilde{O}(BM^{\frac{3}{4}}K^{\frac{3}{4}}\log T\sqrt{T})$$

*where $B$ is the batch size, $M$ is the number of candidate algorithms, $K$ is the number of classes, and $T$ is the number of rounds.*

We omit the proof in this paper and would like to point the readers to section 6.2.1 in Russo and Van Roy [2014] for the proof sketch. As mentioned in the paper, detailed confidence ellipsoid of least squares estimate and ellipsoid radius upper bound can be recovered from pages 14-15 of Abbasi-Yadkori et al. [2011].

We now present an upper bound on the Bayesian regret of `TAILOR` , which utilizes standard sub-Gaussian tail bounds based on observations of $y^{t,j}$ instead of confidence ellipsoids derived from only observing reward signals of $r^{t,j}$.

**Theorem 5.2** (Proof in Appendix B). *The Bayesian regret of `TAILOR` is bounded by*

$$BR(\text{TAILOR}) \leq O(B\sqrt{MT(\log T + \log M)})$$

*where $B$ is the batch size, $M$ is the number of candidate algorithms and $T$ is total number of rounds.*

We delay our complete proof to Appendix B. To highlight the key difference of our analysis from Russo and Van Roy [2014], their algorithm only rely on observations of $r^{t,j}$ for each round $t \in [T]$ and element $j \in [B]$ in a batch. To estimate $\theta^1, ..., \theta^M$, they use the least squares estimator to form confidence ellipsoids. In our analysis, we utilize observations of $y$'s up to round $t$ and form confidence intervals directly around each of $\langle v^t, \theta^1\rangle, ..., \langle v^t, \theta^M\rangle$ by unbiased estimates $\{\langle v^t, y\rangle$.

## 5.1 Time Complexity

Let $N_{train}$ denote the total neural network training. The time complexity of collecting each batch for each active learning algorithm $\mathcal{A}_i$ can be separated into $P_i$ and $Q_i$, which are the computation complexity for preprocessing and selection of each example respectively. As examples of preprocessing, BADGE [Ash et al., 2019] computes gradient embeddings, SIMILAR [Kothawade et al., 2021] further also compute similarity kernels, GALAXY [Zhang et al., 2022] constructs linear graphs, etc. The selection complexities are the complexities of each iteration of K-means++ in BADGE, greedy submodular optimization in SIMILAR, and shortest shortest path computation in GALAXY. Therefore, for any individual algorithm $\mathcal{A}_i$, the computation complexity is then

| DATASET | $K$ | $N$ | CLASS IMB RATIO | BINARY IMB RATIO |
|---|---|---|---|---|
| CELEBA [LIU ET AL., 2018] | 40 | 162770 | .0273 | .2257 |
| COCO [LIN ET AL., 2014] | 80 | 82081 | .0028 | .0367 |
| VOC [EVERINGHAM ET AL., 2010] | 20 | 10000 | .0749 | .0721 |
| CAR [KRAUSE ET AL., 2013] | 10 | 12948 | .1572 | .1200 |
| IMAGENET [DENG ET AL., 2009] | 1000 | 1281167 | .5631 | — |
| KUZUSHIJI-49 [CLANUWAT ET AL., 2018] | 49 | 23236 | .0545 | — |
| CALTECH256 [GRIFFIN ET AL., 2007] | 256 | 24486 | .0761 | — |
| IMB CIFAR-10 [KRIZHEVSKY ET AL., 2009] | 2 | 50000 | .1111 | — |
| IMB CIFAR-100 [KRIZHEVSKY ET AL., 2009] | 10 | 50000 | .0110 | — |
| IMB SVHN [NETZER ET AL., 2011] | 2 | 73257 | .0724 | — |

Table 1: Details for multi-label and multi-class classification datasets. $K$ and $N$ denote the number of classes and pool size respectively. *Class Imbalance Ratio* represents the class imbalance ratio between the smallest and the largest class. We also report *Binary Imbalance Ratio* for multi-label datasets, which is defined as the average positive ratio over classes, i.e., $\frac{1}{K}\sum_{i\in[K]}(N_i/N)$ where $N_i$ denotes the number of examples in class $i$.

$O(N_{train} + TP_i + TBQ_i)$ where $T$ is the total number of rounds and $B$ is the batch size. When running TAILOR, as we do not know which algorithms are selected, we provide a worst case upper bound of $O(N_{train} + T \cdot (\sum_{i=1}^{M} P_i) + TB \cdot \max_{i\in[M]} Q_i)$, where the preprocessing is done for every candidate algorithm. In practice, some of the preprocessing operations such as gradient embedding computation could be shared among multiple algorithms, thus only need to be computed once. As a practical note in all of our experiments, TAILOR is more than 20% faster than the slowest candidate algorithm as it selects a diverse set of candidate algorithms instead of running a single algorithm the entire time. Also, the most significant time complexity in practice often lies in neural network retraining. The retraining time dominates the running time of all algorithms including reward and Thompson sampling complexities.

# 6 Experiments

In this section, we present results of TAILOR in terms of classification accuracy, class-balance of collected labels, and total number of positive examples for multi-label active search. Motivated by the observations, we also propose some future directions at the end.

## 6.1 Setup

**Datasets.** Our experiments span ten datasets with class-imbalance as shown in Table 1. For multi-label experiments, we experiment on four datasets including CelebA, COCO, VOC and Stanford Car datasets. While the Stanford Car dataset is a multi-class classification dataset, we transform it into a multi-label dataset as detailed in Appendix A.3. For multi-class classification datasets, ImageNet, Kuzushiji-49 and Caltech256 are naturally unbalanced datasets, while CIFAR-10 with 2 classes, CIFAR-100 with 10 classes and SVHN with 2 classes are derived from the original dataset following Zhang et al. [2022]. Specifically, we keep the first $K-1$ classes from the original dataset and treat the rest of the images as a large out-of-distribution $K$-th class.

**Implementation Details.** We conduct experiments on varying batch sizes anywhere from $B = 500$ to $B = 10000$. To mirror a limited training budget [Citovsky et al., 2021, Emam et al., 2021], we allow 10 or 20 batches in total for each dataset, making it extra challenging for our adaptive algorithm selection due to the limited rounds of interaction.

Moreover, we assumed observations of $y$ are sampled from stationary distributions $\mathbb{P}_{\theta^1}, ..., \mathbb{P}_{\theta^M}$ in our analysis. However, these distributions could be dynamically changing. In our implementation, we use a simple trick to discount the past observations, where we change the posterior update in Algorithm 2 to $a^i \leftarrow \gamma a^i + \sum_{j:\alpha^{t,j}=i} y^{t,j}$ and $b^i \leftarrow \gamma b^i + \sum_{j:\alpha^{t,j}=i}(1 - y^{t,j})$. We set the discounting factor $\gamma$ to be .9 across all experiments. As will be discussed in Section 6.3, we find non-stationarity in

$\{\mathbb{P}_{\theta^k}\}_{k=1}^M$ an interesting future direction to study. Lastly, we refer the readers to Appendix A for additional implementation details.

**Baseline Algorithms.** In our experiments, we choose a representative and popular subset of the deep AL algorithms and active search algorithms discussed in Section 2 as our baselines. To demonstrate the ability of `TAILOR`, number of candidate algorithms $M$ ranges from tens to hundreds for different datasets. The baselines can be divided into three categories:

- We include off-the-shelf active learning and active search algorithms such as **EMAL** [Wu et al., 2014] and **Weak Supervision** [Ranganathan et al., 2018] for multi-label classification and **Confidence sampling** [Settles, 2009], **BADGE** [Ash et al., 2019], **Modified Submodular** optimization motivated by Kothawade et al. [2021] for multi-class classification. More implementation details can be found in Appendices A.1 and A.2.
- We derive individual candidate algorithms based on a per-class decomposition [Boutell et al., 2004]. For most likely positive sampling [Warmuth et al., 2001, 2003, Jiang et al., 2018], an active search strategy and abbreviated as **MLP**, we obtain $K$ algorithms where the $i$-th algorithm selects examples most likely to be in the $i$-th class.
  For multi-label classification, we also include $K$ individual **GALAXY** algorithms [Zhang et al., 2022] and $K$ **Uncertainty sampling** algorithms. To further elaborate, the original **GALAXY** work by Zhang et al. [2022] construct $K$ one-vs-rest linear graphs, one for each class. **GALAXY** requires finding the shortest shortest path among all $K$ graphs, an operation whose computation scales linearly in $K$. When $K$ is large, this becomes computationally prohibitive to run. Therefore, we instead include $K$ separate GALAXY algorithms, each only bisecting on one of the one-vs-rest graphs. This is equivalent with running $K$ **GALAXY** algorithms, one for each binary classification task between class $i \in [K]$ and the rest.
  For **Uncertainty sampling** in multi-label settings, we similarly have $K$ individual uncertainty sampling algorithms, where the $i$-th algorithm samples the most uncertain example based only on the binary classification task of class $i$.
  As baselines for each type of algorithms above, we simply interleave the set of $K$ algorithms uniformly at random.
- We compare against other adaptive meta selection algorithms, including **Random Meta** which chooses candidate algorithms uniform at random and **ALBL Sampling** [Hsu and Lin, 2015]. The candidate algorithms include all of the active learning baselines. In Appendix C, we also provide an additional study of including active search baselines as candidate algorithms.

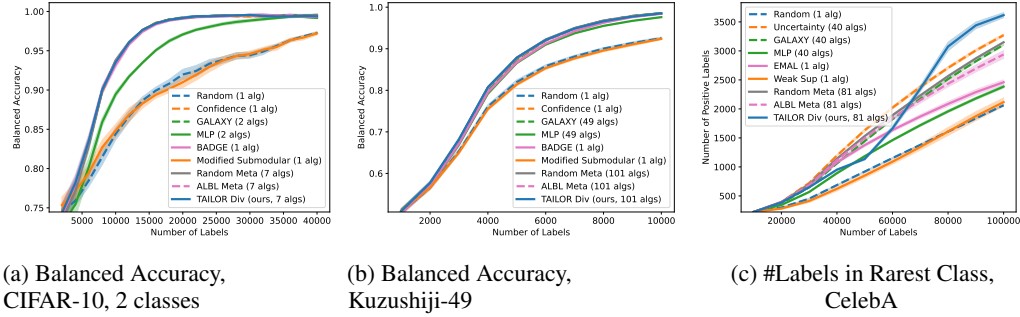

(a) Balanced Accuracy,
CIFAR-10, 2 classes

(b) Balanced Accuracy,
Kuzushiji-49

(c) #Labels in Rarest Class,
CelebA

Figure 3: Performance of `TAILOR` against baselines on selected settings. (a) and (b) shows accuracy metrics of the algorithms. (c) shows class-balancedness of labeled examples. All performances are averaged over four trials with standard error plotted for each algorithm. The curves are smoothed with a sliding window of size 3.

## 6.2 Results

**Multi-class and Multi-label Classification.** For evaluation, we focus on `TAILOR`'s comparisons against both existing meta algorithms and the best baseline respectively. In all classification experiments, `TAILOR` uses the class diversity reward in Section 4.1. For accuracy metrics, we utilize mean average precision for multi-label classification and balanced accuracy for multi-class classification. As a class diversity metric, we look at the size of the smallest class based on collected labels. All experiments are measured based on active annotation performance over the pool [Zhang et al., 2022].

As shown in Figures 2, 3 and Appendix D, when comparing against existing meta algorithms, `TAILOR` performs better on all datasets in terms of both accuracy and class diversity metrics. **ALBL sampling** performs similar to **Random Meta** in all datasets, suggesting the ineffectiveness of training accuracy based rewards proposed in Hsu and Lin [2015] and Pang et al. [2018]. When comparing against the best baseline algorithm, `TAILOR` performs on par with the best baseline algorithm on nine out of ten datasets in terms of accuracy and on all datasets in terms of class diversity. On the CelebA dataset, `TAILOR` even outperforms the best baseline by significant margin in accuracy. As discussed in Appendix E, `TAILOR` achieves this by selecting a *combination* of other candidate algorithm instead of choosing only the best baseline. On four out of the ten datasets, `TAILOR` outperforms the best baseline in class diversity. Collectively, this shows the power of `TAILOR` in identifying the best candidate algorithms over different dataset scenarios. Moreover in Appendix D.4, we conduct an ablation study of the accuracy of the rarest class (determined by the ground truth class distribution). `TAILOR` significantly outperform baselines suggesting its advantage in improving the accuracy on *all* classes. Lastly, shown in Appendix E, we also find `TAILOR` selects algorithms more aggressively than existing meta algorithms. The most frequent algorithms also align with the best baselines.

On the other hand for the Caltech256 dataset shown in Figure 16, `TAILOR` under-performs **confidence sampling** in terms of accuracy. We conjecture this is because the larger classes may not have sufficient examples and have much space for improvement before learning the smaller classes. Nevertheless, `TAILOR` was able to successfully collect a much more class-diverse dataset while staying competitive to other baseline methods.

**Multi-label Search.** We use the multi-label search reward proposed in Section 4.1. As shown in Figure 4 and Appendix D.3, on three of the four datasets, `TAILOR` performs better than the best baseline algorithm in terms of total collected positive labels. On the fourth dataset, `TAILOR` performs second to and on par with the best baseline. This shows `TAILOR`'s ability in choosing the best candidate algorithms for active search.

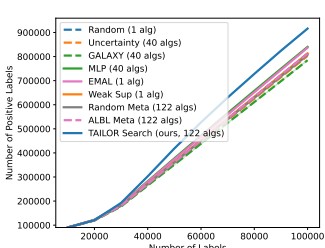

Figure 4: Total positive labels for active search, CelebA

### 6.3 Future Work

While our experiments focus on class-imbalanced settings, `TAILOR`'s effectiveness on balanced datasets warrants future study through further experiments and alternative reward design. We also find studying non-stationarity in label distributes $\{\mathbb{P}_{\theta_i}\}_{i=1}^{M}$ an interesting next step.

## 7 Choosing Candidate Algorithms

Our paper proposes an adaptive selection procedure over candidate deep AL algorithms. When judging individual deep AL algorithms, current standards in the research community tend to focus on whether an algorithm performs well on *all* dataset and application instances. However, we see value in AL algorithms that perform well only in certain instances. Consider, for example, an AL algorithm that performs well on 25% of previous applications, but poorly on the other 75%. One may wish to include this algorithm in `TAILOR` because the new application might be similar to those where it performs well. From the perspective of `TAILOR` , a "good" AL algorithm need not perform well on all or even most of a range of datasets, it just needs to perform well on a significant number of datasets.

On the other hand, as suggested by our regret bound that scales with $M$, one should not include too many algorithms. In fact, there are exponential number of possible AL algorithms, which could easily surpass our labeling budget and overwhelm the meta selection algorithm. In practice, one could leverage extra information such as labeling budget, batch size and model architecture to choose proper set of candidate algorithms to target their settings.

## Acknowledgement

We would like to thank Aude Hofleitner and Shawndra Hill for the support of this research project, and Kevin Jamieson, Yifang Chen and Andrew Wagenmaker for insightful discussions. Robert Nowak would like to thank the support of NSF Award 2112471.

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

# A Implementation Details

## A.1 Deep Active Learning Decomposition

For any uncertainty sampling algorithm, picking the top-$B$ most uncertain examples can be easily decomposed into an iterative procedure that picks the next most uncertain example. Next, for diversity based deep active learning algorithms, one usually rely on a greedy iterative procedure to collect a batch, e.g. K-means++ for BADGE [Ash et al., 2019] and greedy K-centers for Coreset [Sener and Savarese, 2017]. Lastly, deep active learning algorithms such as Cluster-Margin [Citovsky et al., 2021] and GALAXY [Zhang et al., 2022] have already proposed their algorithms as iterative procedures that select unlabeled examples sequentially.

## A.2 Implementation of Modified Submodular

Instead of requiring access to a balanced holdout set [Kothawade et al., 2021], we construct the balanced set using training examples. We use the Submodular Mutual Information function FLQMI as suggested by Table 1 of Kothawade et al. [2021]. The proposed greedy submodular optimization is itself an iterative procedure that selects one example at a time. While SIMILAR usually performs well, our modification that discards the holdout set is unfortunately ineffective in our experiments. This is primarily due to the lack of the holdout examples, which may often happen in practical scenarios.

## A.3 Stanford Car Multi-label Dataset

We transform the original labels into 10 binary classes of

1. If the brand is "Audi".
2. If the brand is "BMW".
3. If the brand is "Chevrolet".
4. If the brand is "Dodge".
5. If the brand is "Ford".
6. If the car type is "Convertible".
7. If the car type is "Coupe".
8. If the car type is "SUV".
9. If the car type is "Van".
10. If the car is made in or before 2009.

## A.4 Negative Weighting for Common Classes

For multi-label classifications, for some classes, there could be more positive associations (label of 1s) than negative associations (label of 0s). Therefore, in those classes, the rarer labels are negative. In class diverse reward $\langle v_{div}^t, y \rangle$ in Section 4.1, we implement an additional weighting of $\mathbb{1}_{rare}^t * v_{div^t}$, where $*$ denotes an elementwise multiplication. Here, each element $\mathbb{1}_{rare,i}^t \in \{1, -1\}$ takes value $-1$ when $\text{COUNT}^t(i)$ is larger than half the size of labeled set. This negative weighting can been seen as upsampling negative class associations when positive associations are the majority.

## A.5 Model Training

All of our experiments are conducted using the ResNet-18 architecture [He et al., 2016] pretrained on ImageNet. We use the Adam optimizer [Kingma and Ba, 2014] with learning rate of 1e-4 and weight decay of 5e-5.

# B Proof of Theorem 5.2

Our proof follows a similar procedure from regret analysis for Thompson Sampling of the stochastic multi-armed bandit problem [Lattimore and Szepesvári, 2020]. Let $\alpha^t := \{\alpha^{t,j}\}_{j=1}^B$ and $y^t :=$

$\{y^{t,j}\}_{j=1}^{B}$ denote the actions and observations from the $i$-th round. We define the history up to $t$ as $H_t = \{\alpha^1, y^1, \alpha^2, y^2, ..., \alpha^{t-1}, y^{t-1}\}$. Moreover, for each $i \in [M]$, we define $H_{t,i} = \{y^{t',j} \in H_t : \alpha^{t',j} = i\}$ as the history of all observations made by choosing the $i$-th arm (algorithm).

Now we analyze reward estimates at each round $t$. When given history $H_t$ and arm $i \in [M]$, each observation $y \in H_{t,i}$ is an unbiased estimate of $\theta^i$ as $y \sim \mathbb{P}_{\theta^i}$. Therefore, for any fixed $v^t$, $\langle v^t, y \rangle$ is an unbiased estimate of the expected reward $\langle v^t, \theta^i \rangle$, which we denote by $\mu^{t,i}$.

For each arm $i$, we can then obtain empirical reward estimate $\bar{\mu}^{t,i}$ of the true expected reward $\mu^{t,i}$ by $\bar{\mu}^{t,i} := \frac{1}{1 \vee |H_{t,i}|} \sum_{y \in H_{t,i}} \langle v^t, y \rangle$ where $\bar{\mu}^{t,i} = 0$ if $|H_{t,i}| = 0$. Since expected rewards and reward estimates are bounded by $[-1, 1]$, by standard sub-Gaussian tail bounds, we can then construct confidence interval,

$$\mathbb{P}\left(\forall i \in [M], t \in [T], |\bar{\mu}^{t,i} - \mu^{t,i}| \le d^{t,i}\right) \ge 1 - \frac{1}{T}$$

where $d^{t,i} := \sqrt{\frac{8 \log(MT^2)}{1 \vee |H_{t,i}|}}$. Additionally, we define upper confidence bound as $U^{t,i} = \text{clip}_{[-1,1]}\left(\bar{\mu}^{t,i} + d^{t,i}\right)$.

At each iteration $t$, we have the posterior distribution $\mathbb{P}(\Theta = \cdot | H_t)$ of the ground truth $\Theta = \{\theta^i\}_{i=1}^{M}$. $\widehat{\Theta} = \{\widehat{\theta}^i\}_{i=1}^{M}$ is sampled from this posterior. Consider $i_\star^t = \arg\max_{i \in M} \langle v^t, \theta^i \rangle$ and $\alpha^{t,j} = \arg\max_{i \in M} \langle v^t, \widehat{\theta}^i \rangle$. The distribution of $i_\star^t$ is determined by the posterior $\mathbb{P}(\Theta = \cdot | H_t)$. The distribution of $\alpha^{t,j}$ is determined by the distribution of $\widehat{\Theta}$, which is also $\mathbb{P}(\Theta = \cdot | H_t)$. Therefore, $i_\star^t$ and $\alpha^{t,j}$ are identically distributed. Furthermore, since the upper confidence bounds are deterministic functions of $i$ when given $H_t$, we then have $\mathbb{E}[U^{t,\alpha^{t,j}} | H_t] = \mathbb{E}[U^{t,i_\star^t} | H_t]$.

As a result, we upper bound the Bayesian regret by

$$BR(\texttt{TAILOR}) = \mathbb{E}\left[\sum_{t=1}^{T} \sum_{j=1}^{B} \mu^{t,i_\star^t} - \mu^{t,\alpha^{t,j}}\right]$$

$$= \mathbb{E}\left[\sum_{t=1}^{T} \sum_{j=1}^{B} (\mu^{t,i_\star^t} - U^{t,i_\star^t}) + (U^{t,\alpha^{t,j}} - \mu^{t,\alpha^{t,j}})\right].$$

Now, note that since $\bar{\mu}^{t,i} \in [-1,1]$ we have $\text{clip}_{[-1,1]}\left(\bar{\mu}^{t,i} + d^{t,i}\right) = \text{clip}_{[-\infty,1]}\left(\bar{\mu}^{t,i} + d^{t,i}\right)$, where only the upper clip takes effect. Based on the sub-Gaussian confidence intervals $\mathbb{P}\left(\forall i \in [M], t \in [T], |\bar{\mu}^{t,i} - \mu^{t,i}| \le d^{t,i}\right) \ge 1 - \frac{1}{T}$, we can derive the following two confidence bounds:

$$\mathbb{P}(\forall i \in [M], t \in [T], \mu^{t,i} > U^{t,i}) = \mathbb{P}(\forall i \in [M], t \in [T], \mu^{t,i} > \text{clip}_{[-1,1]}(\bar{\mu}^{t,i} + d^{t,i}))$$

$$= \mathbb{P}(\forall i \in [M], t \in [T], \mu^{t,i} > \bar{\mu}^{t,i} + d^{t,i}), \text{ since } \mu^{t,i} \le 1$$

$$= \mathbb{P}(\forall i \in [M], t \in [T], \mu^{t,i} - \bar{\mu}^{t,i} > d^{t,i}) \le \frac{1}{2T}$$

$$\mathbb{P}(\forall i \in [M], t \in [T], U^{t,i} - \mu^{t,i} > 2d^{t,i}) = \mathbb{P}(\forall i \in [M], t \in [T], \text{clip}_{[-1,1]}(\bar{\mu}^{t,i} + d^{t,i}) - \mu^{t,i} > 2d^{t,i})$$

$$\le \mathbb{P}(\forall i \in [M], t \in [T], \bar{\mu}^{t,i} + d^{t,i} - \mu^{t,i} > 2d^{t,i})$$

$$= \mathbb{P}(\forall i \in [M], t \in [T], \bar{\mu}^{t,i} - \mu^{t,i} > d^{t,i}) \le \frac{1}{2T}.$$

Now with the decomposition,

$$BR(\texttt{TAILOR}) = \mathbb{E}\left[\sum_{t=1}^{T} \sum_{j=1}^{B} \mu^{t,i_\star^t} - \mu^{t,\alpha^{t,j}}\right]$$

$$= \mathbb{E}\left[\sum_{t=1}^{T} \sum_{j=1}^{B} \mu^{t,i_\star^t} - U^{t,i_\star^t}\right] + \mathbb{E}\left[\sum_{t=1}^{T} \sum_{j=1}^{B} U^{t,\alpha^{t,j}} - \mu^{t,\alpha^{t,j}}\right]$$

we can bound the two expectations individually.

First, to bound $\mathbb{E}\left[\sum_{t=1}^{T}\sum_{j=1}^{B}\mu^{t,i_\star^t} - U^{t,i_\star^t}\right]$, we note that $\mu^{t,i_\star^t} - U^{t,i_\star^t}$ is negative with high probability. Also, the maximum value this can take is bounded by 2 as $\mu^{t,i}, U^{t,i} \in [-1,1]$. Therefore, we have

$$\mathbb{E}\left[\sum_{t=1}^{T}\sum_{j=1}^{B}\mu^{t,i_\star^t} - U^{t,i_\star^t}\right] \leq \left(\sum_{t=1}^{T}\sum_{j=1}^{B}0 \cdot \mathbb{P}(\mu^{t,i_\star^t} <= U^{t,i_\star^t}) + 2 \cdot \mathbb{P}(\mu^{t,i_\star^t} > U^{t,i_\star^t})\right) \leq 2TB \cdot \frac{1}{2T} = B.$$

Next, to bound $\mathbb{E}\left[\sum_{t=1}^{T}\sum_{j=1}^{B}U^{t,\alpha^{t,j}} - \mu^{t,\alpha^{t,j}}\right]$ we decompose it similar to the above:

$$\mathbb{E}\left[\sum_{t=1}^{T}\sum_{j=1}^{B}U^{t,\alpha^{t,j}} - \mu^{t,\alpha^{t,j}}\right] \leq \left(\sum_{t=1}^{T}\sum_{j=1}^{B}2\mathbb{P}(U^{t,\alpha^{t,j}} - \mu^{t,\alpha^{t,j}} > 2d^{t,i})\right) + \left(\sum_{t=1}^{T}\sum_{j=1}^{B}2d^{t,i}\right)$$

$$\leq B + \left(\sum_{t=1}^{T}\sum_{j=1}^{B}\sqrt{\frac{32\log(MT^2)}{1 \vee |H_{t,\alpha^{t,j}}|}}\right)$$

where recall that $|H_{t,i}|$ is the number of samples collected using algorithm $i$ in rounds $\leq t$.

To bound the summation, we utilize the fact that $\frac{1}{1\vee|H_{t,i}|} \leq \frac{B}{k}$ for each $k \in [|H_{t,i}|, |H_{t+1,i}|]$, since $|H_{t+1,i}| - |H_{t,i}| \leq B$. As a result, we get

$$\sum_{t=1}^{T}\sum_{j=1}^{B}\sqrt{\frac{32\log(MT^2)}{1 \vee |H_{t,\alpha^{t,j}}|}}$$

$$\leq \sum_{t=1}^{T}\sum_{i=1}^{M}\sum_{k=1}^{|H_{T,i}|}\sqrt{\frac{32\log(MT^2) \cdot B}{k}}$$

$$\leq O(\sqrt{B(\log T + \log M)})\sum_{i=1}^{M}\sqrt{|H_{T,i}|}$$

$$\leq O(\sqrt{B(\log T + \log M)}) \cdot O(\sqrt{BMT}) = O(B\sqrt{MT(\log T + \log M)})$$

where last two inequalities follow from simple algebra and the fact that $\sum_{i=1}^{M}|H_{T,i}| = TB$.

Finally, to combine all of the bounds above, we get $BR(\texttt{TAILOR}) \leq B + B + O(B\sqrt{MT(\log T + \log M)}) = O(B\sqrt{MT(\log T + \log M)})$.

# C  Study of Candidate Algorithms

We compare the performance when we use the following two sets of candidate algorithms:

1. **Active learning algorithms only:** Uncertainty sampling, GALAXY and EMAL for multi-label classification; Uncertainty sampling, GALAXY and BADGE for multi-class classification.

2. **Active learning and search algorithms:** Uncertainty sampling, GALAXY, MLP, EMAL and Weak Sup for multi-label classification; Uncertainty sampling, GALAXY, MLP, BADGE and Modified Submodular for multi-class classification.

Note Modified Submodular is classified as an active search algorithms since we are using a balanced set of training examples as the conditioning set. We are effectively searching for examples similar to the ones that are annotated in these classes.

As shown in Figures 5 and 6, regardless of the meta algorithm, the performance is better when using active learning algorithms as candidates only. Nonetheless, even with active search algorithms as candidates, `TAILOR` still outperforms other meta active learning algorithms.

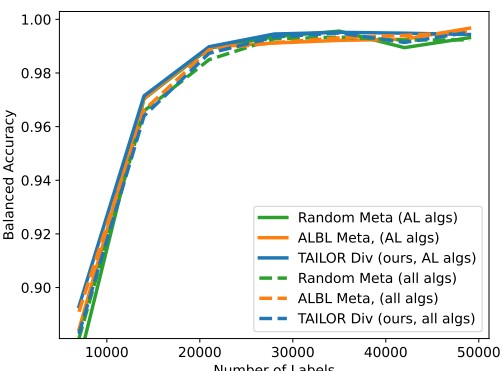

Figure 5: SVHN, Balanced Accuracy

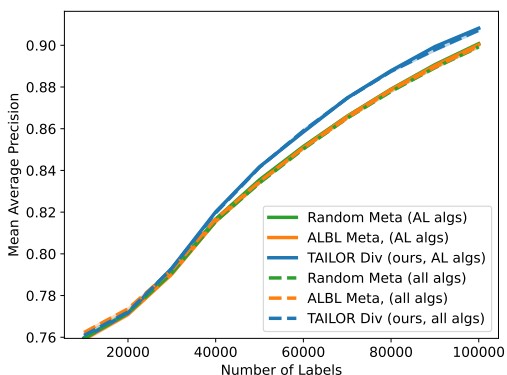

Figure 6: CelebA, mAP

# D   Full Results

All of the results below are averaged from four individual trials except for Imagenet, which is the result of a single trial.

## D.1   Multi-label Classification

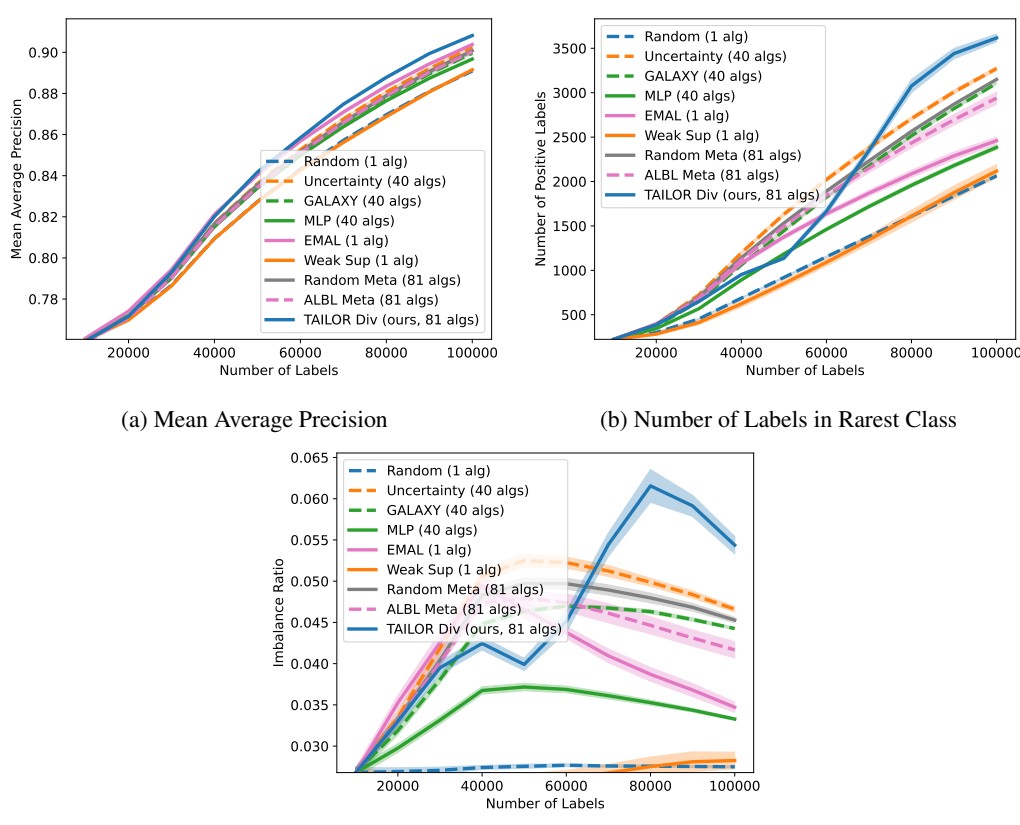

(a) Mean Average Precision

(b) Number of Labels in Rarest Class

(c) Imbalance Ratio

Figure 7: CelebA

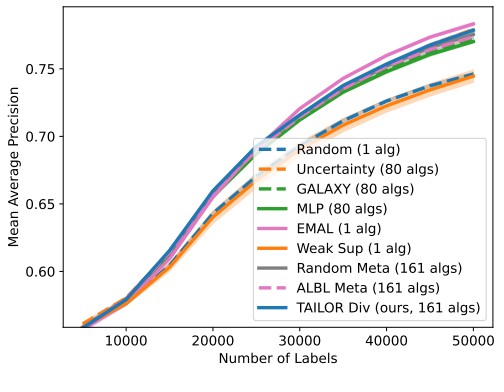

(a) Mean Average Precision

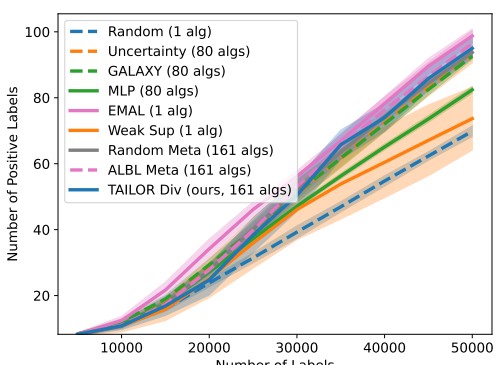

(b) Number of Labels in Rarest Class

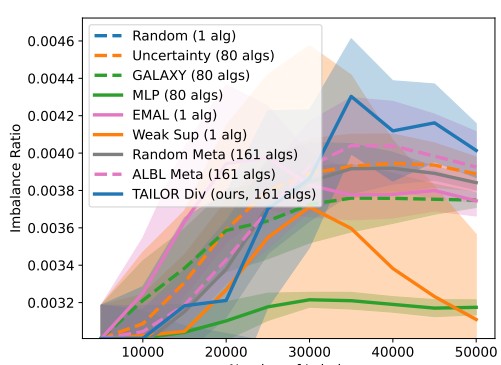

(c) Imbalance Ratio

Figure 8: COCO

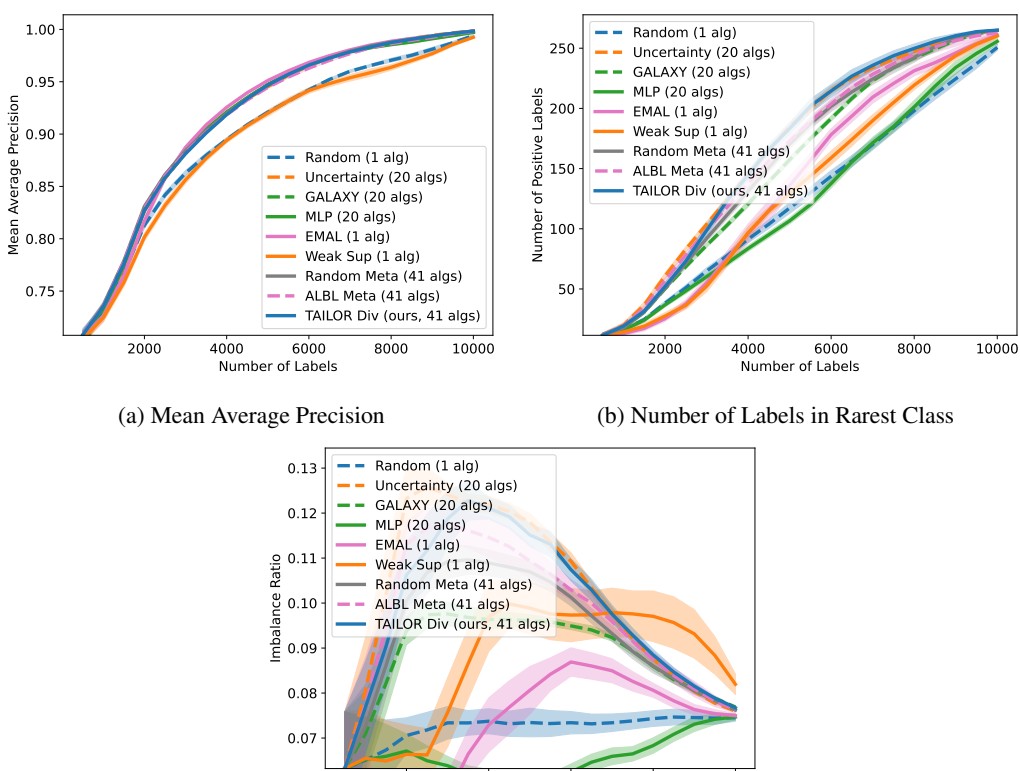

(a) Mean Average Precision

(b) Number of Labels in Rarest Class

(c) Imbalance Ratio

Figure 9: VOC

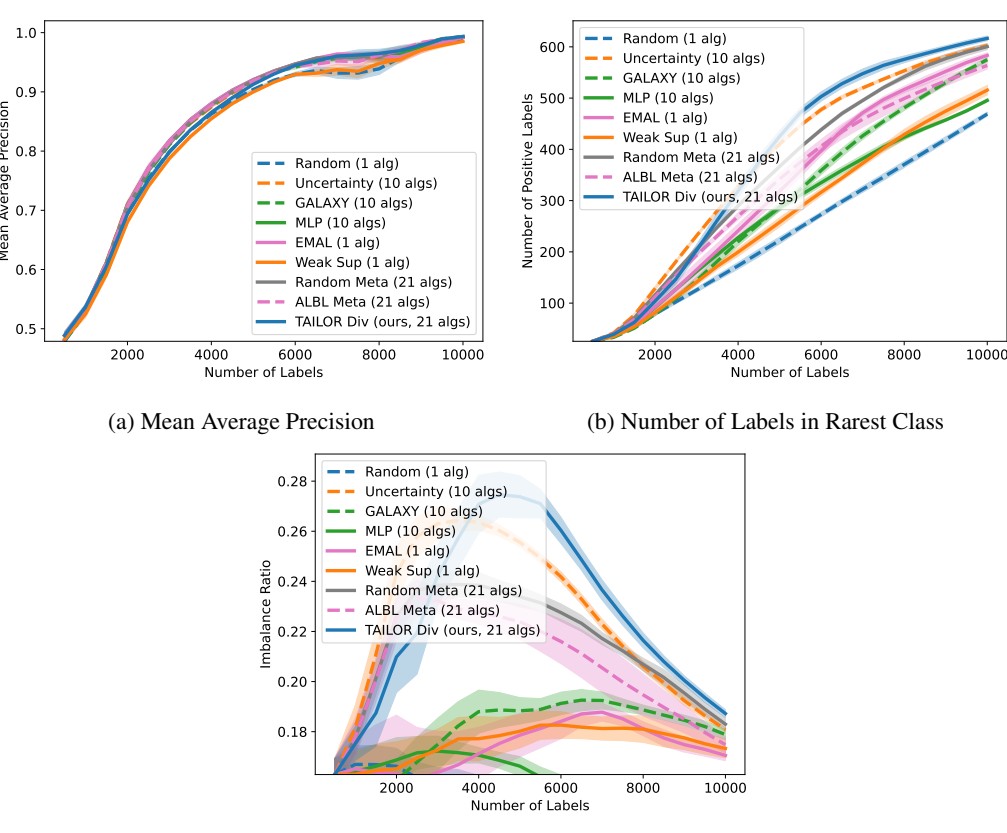

(a) Mean Average Precision

(b) Number of Labels in Rarest Class

(c) Imbalance Ratio

Figure 10: Stanford Car

## D.2 Multi-class Classification

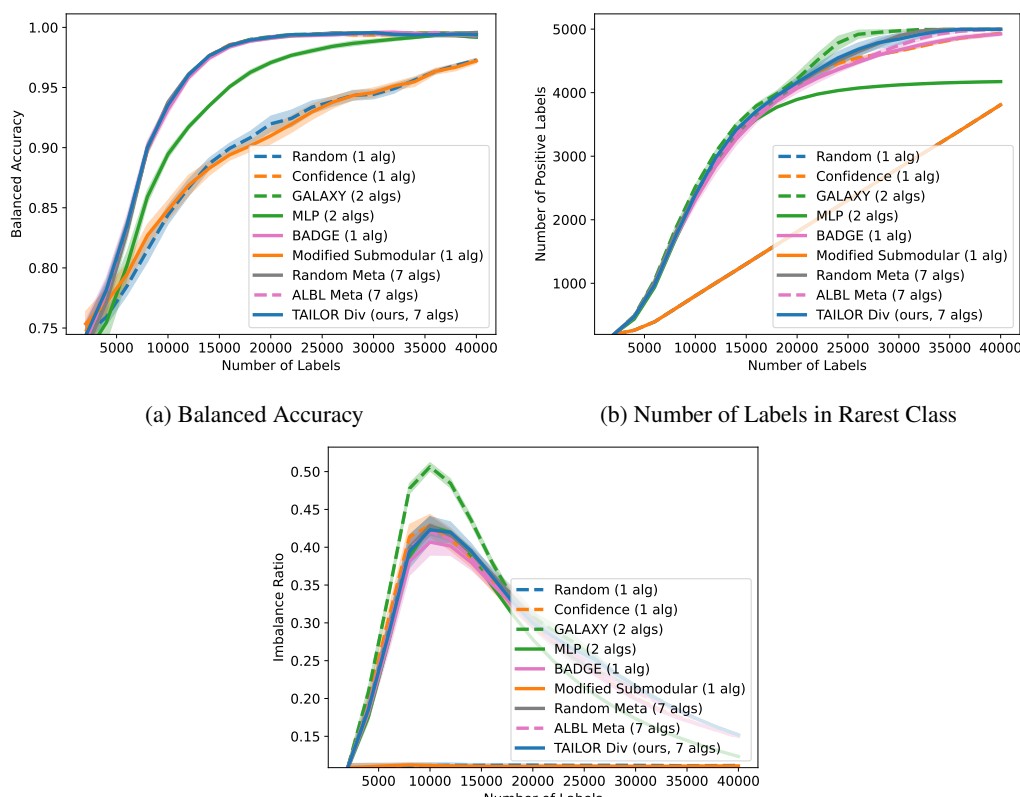

(a) Balanced Accuracy

(b) Number of Labels in Rarest Class

(c) Imbalance Ratio

Figure 11: CIFAR-10, 2 classes

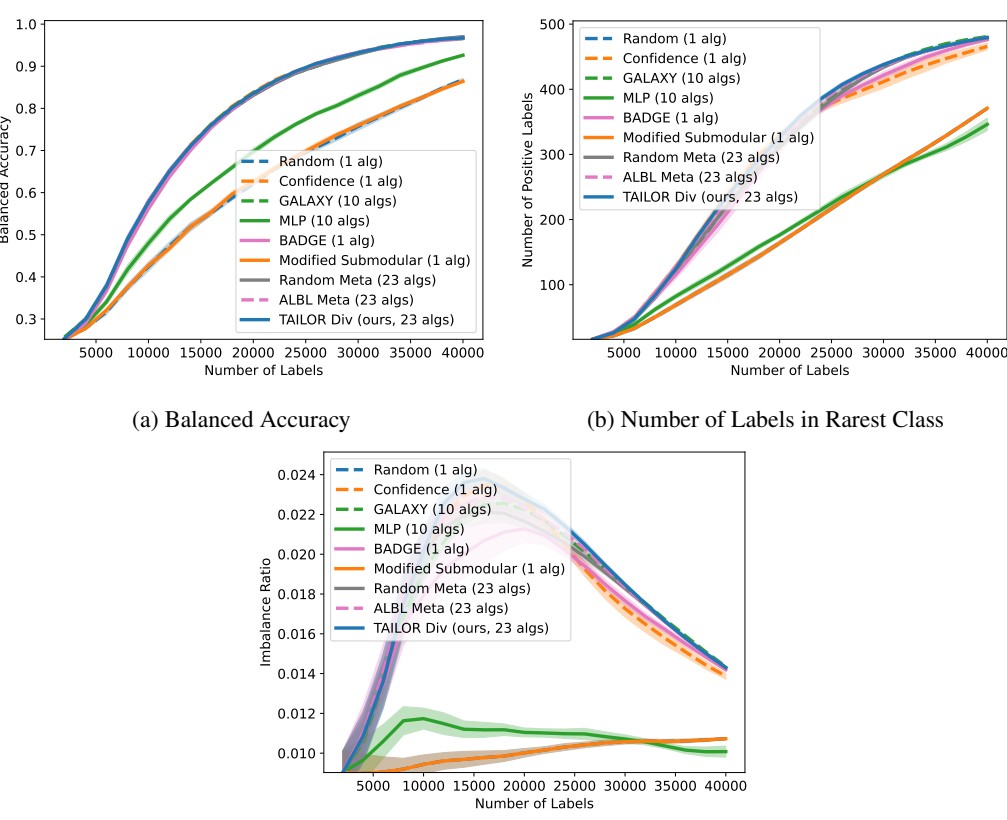

(a) Balanced Accuracy

(b) Number of Labels in Rarest Class

(c) Imbalance Ratio

Figure 12: CIFAR-100, 10 classes

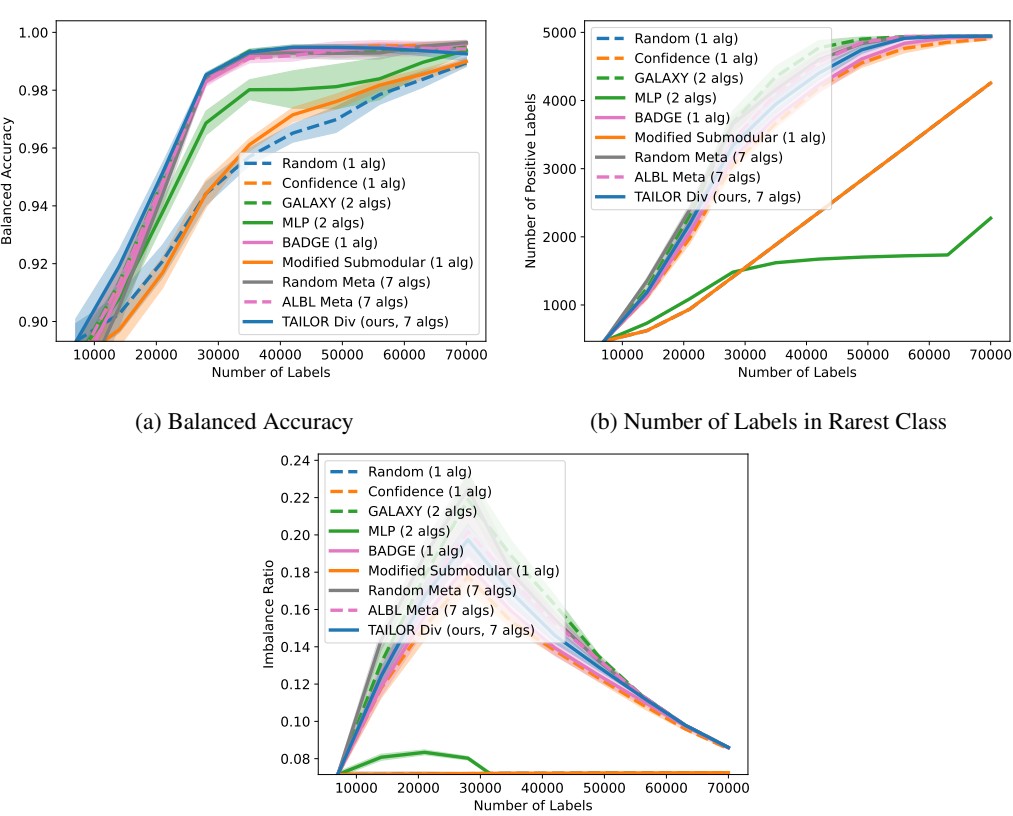

(a) Balanced Accuracy

(b) Number of Labels in Rarest Class

(c) Imbalance Ratio

Figure 13: SVHN, 2 classes

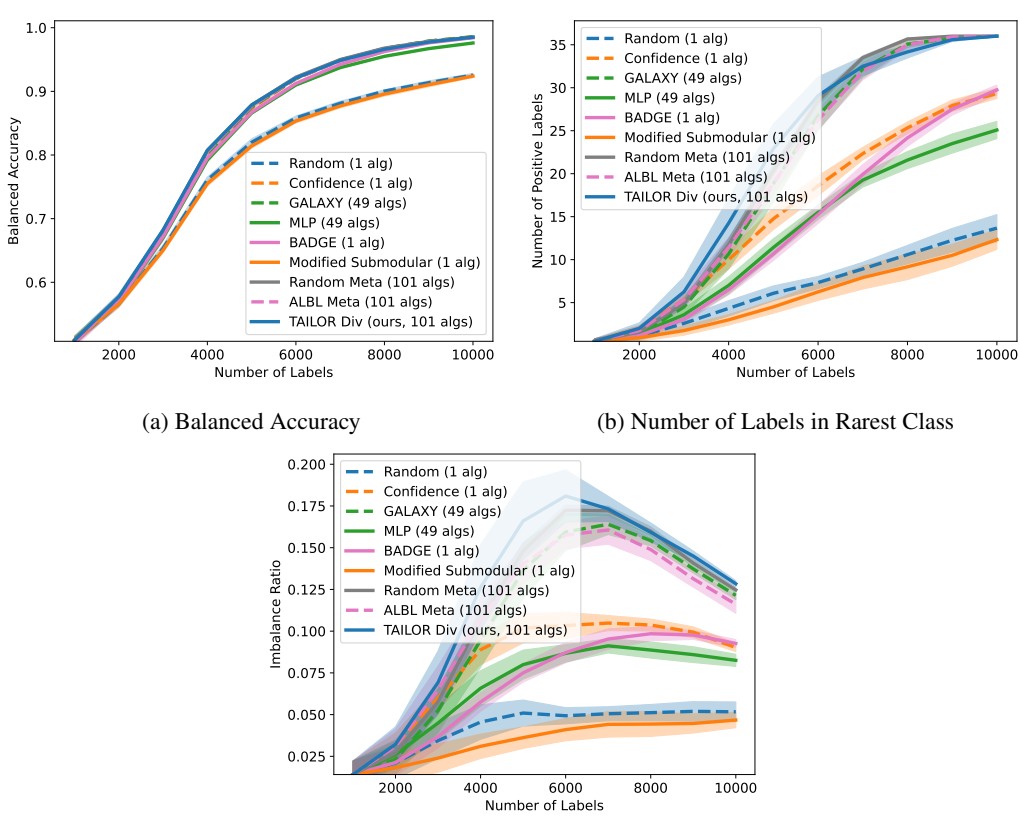

(a) Balanced Accuracy

(b) Number of Labels in Rarest Class

(c) Imbalance Ratio

Figure 14: Kuzushiji-49

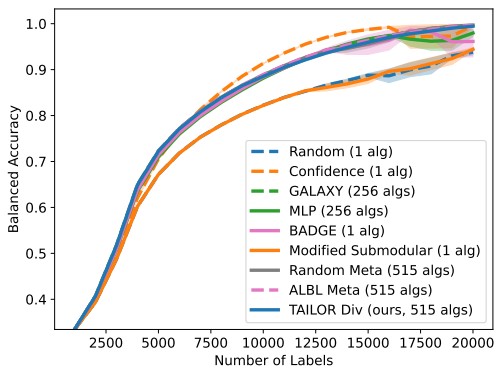

(a) Balanced Accuracy

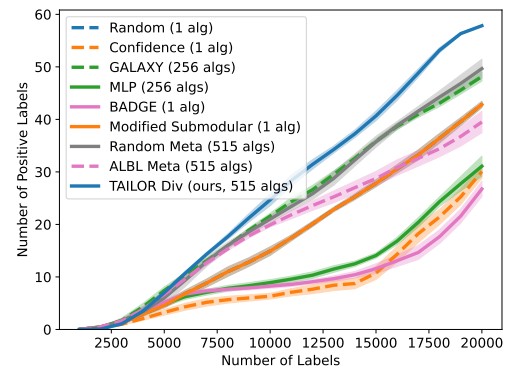

(b) Number of Labels in Rarest Class

Figure 15: Caltech256

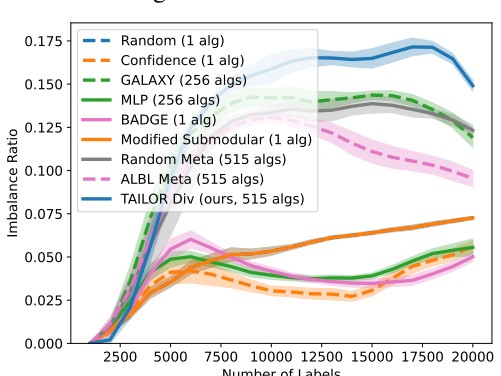

(a) Imbalance Ratio

Figure 16: Caltech256

## D.3 Multi-label Search

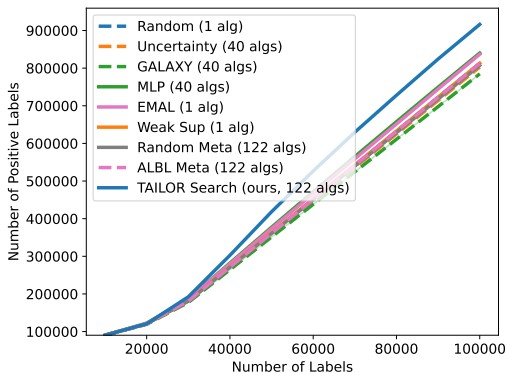

Figure 17: CelebA, Total Number of Positive Labels

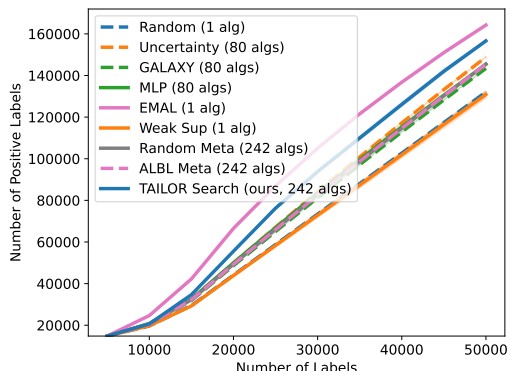

Figure 18: COCO, Total Number of Positive Labels

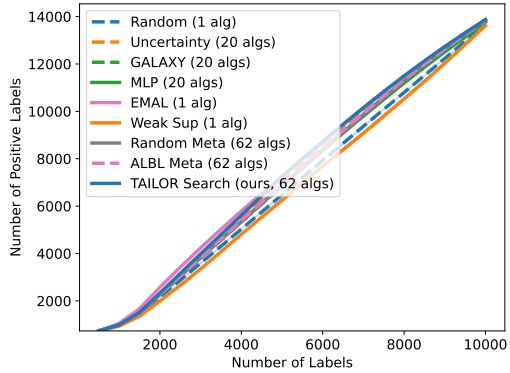

Figure 19: VOC, Total Number of Positive Labels

## D.4 Rarest Class Accuracy

We conduct an ablation study of the accuracy on the rarest class (determined by the ground truth class distribution). TAILOR significantly outperform baselines.

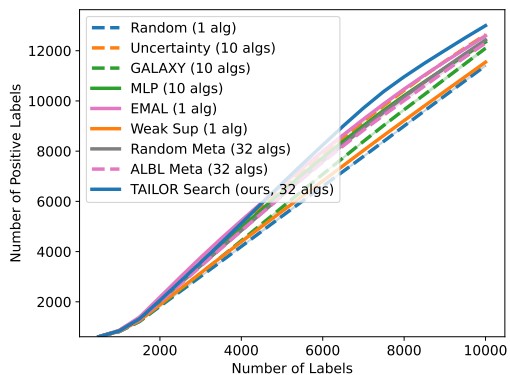

Figure 20: Stanford Car, Total Number of Positive Labels

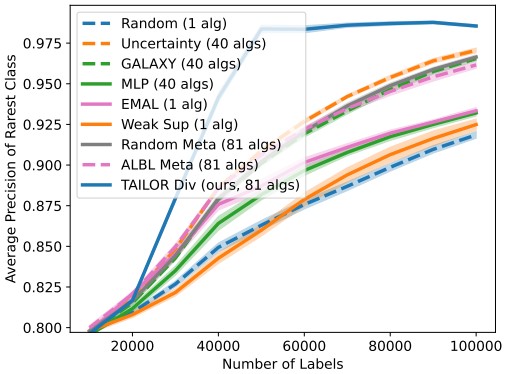

Figure 21: CelebA, Rarest Class Accuracy

# E   What Algorithms Does TAILOR Choose?

In the following two figures, we can see TAILOR chooses a non-uniform set of algorithms to focus on for each dataset. On CelebA, TAILOR out-perform the best baseline, EMAL sampling, by a significant margin. As we can see, TAILOR rely on selecting a *combination* of other candidate algorithms instead of only selecting EMAL.

On the other hand, for the Stanford car dataset, we see TAILOR 's selection mostly align with the baselines that perform well especially in the later phase.

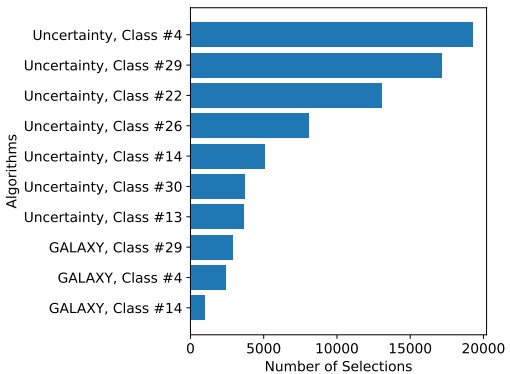

Figure 22: TAILOR Top-10 Most Selected Candidate Algorithms on CelebA Dataset

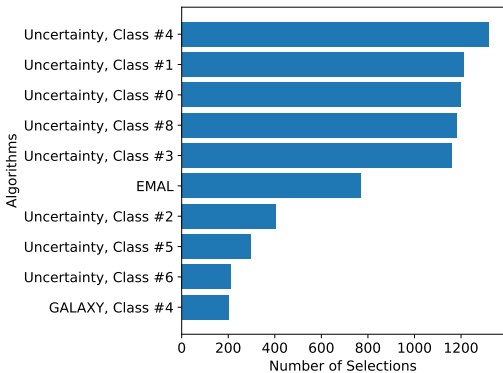

Figure 23: TAILOR Top-10 Most Selected Candidate Algorithms on Stanford Car Dataset

In the following figures, we plot the number of times the most frequent candidate algorithm is chosen. As can be shown, TAILOR chooses candidate algorithm much more aggressively than other meta algorithms in eight out of the ten settings.

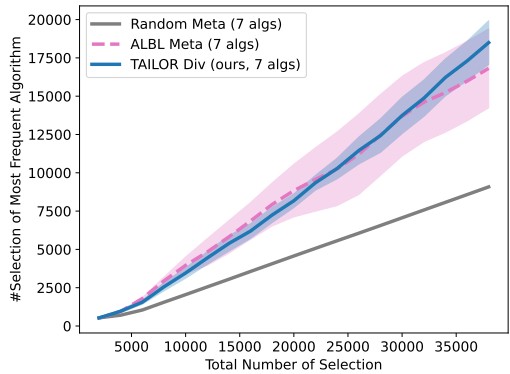

Figure 24: CIFAR-10, 2 Classes, Number of Pulls of The Most Frequent Selection

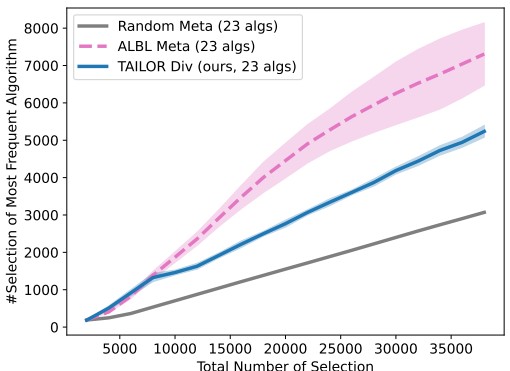

Figure 25: CIFAR-100, 10 Classes, Number of Pulls of The Most Frequent Selection

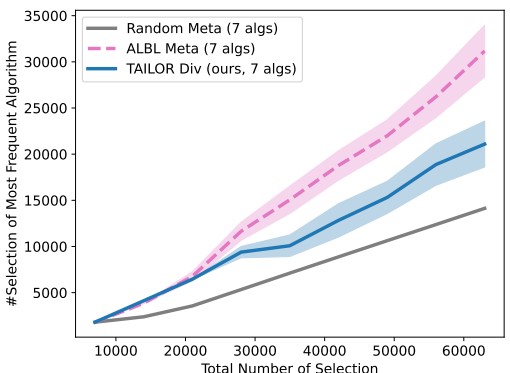

Figure 26: SVHN, 2 Classes, Number of Pulls of The Most Frequent Selection

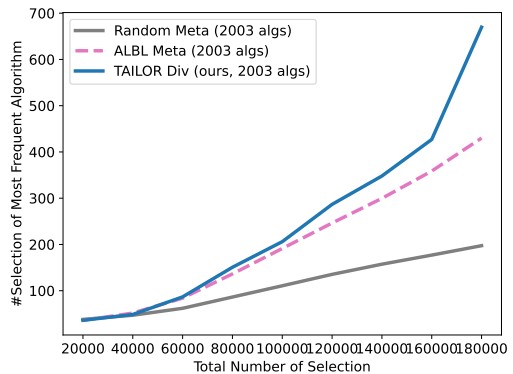

Figure 27: ImageNet-1k, Number of Pulls of The Most Frequent Selection

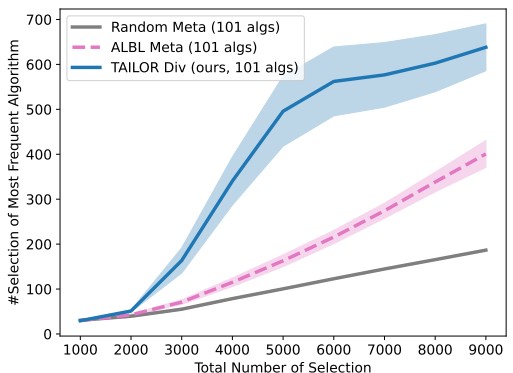

Figure 28: Kuzushiji-49, Number of Pulls of The Most Frequent Selection

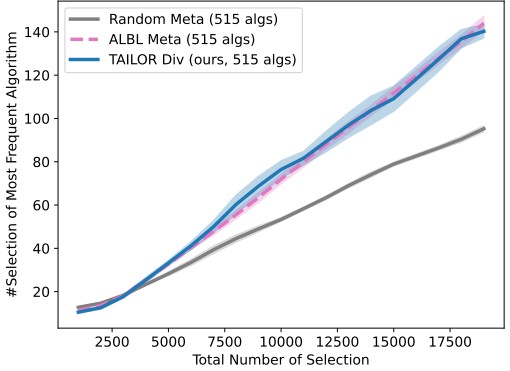

Figure 29: Caltech256, Number of Pulls of The Most Frequent Selection

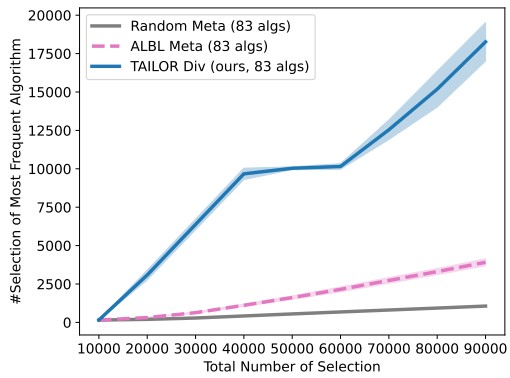

Figure 30: CelebA, Number of Pulls of The Most Frequent Selection

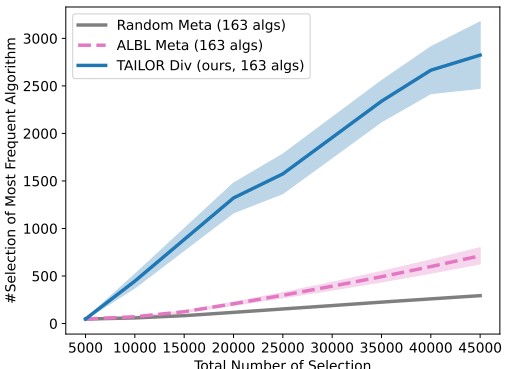

Figure 31: COCO, Number of Pulls of The Most Frequent Selection

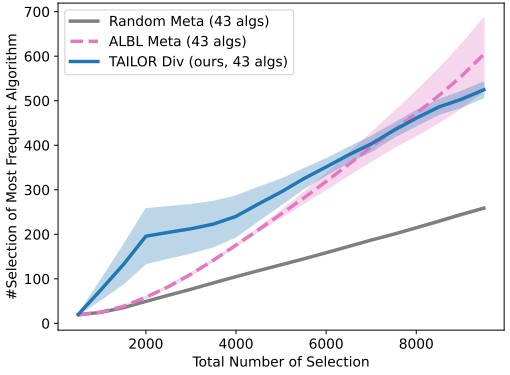

Figure 32: VOC, Number of Pulls of The Most Frequent Selection

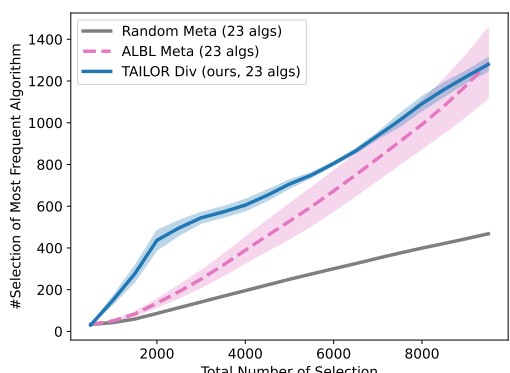

Figure 33: Stanford Car, Number of Pulls of The Most Frequent Selection

