# OpenReview forum: "Algorithm Selection for Deep Active Learning with Imbalanced Datasets"
_NeurIPS.cc/2023/Conference — NeurIPS 2023 poster_

### Official Review · Reviewer_NzMT · 2023-07-01

**Soundness:** 4 excellent
**Presentation:** 4 excellent
**Contribution:** 4 excellent
**Rating:** 7
**Confidence:** 3

**Summary:**

This paper proposes a novel algorithm called TAILOR  that adaptively chooses the active learning algorithm from the candidate set of multiple algorithms.  Novel reward functions are proposed that encourage choosing the algorithm such that both class diversity, as well as informativeness, are maintained while selecting unlabeled samples for the annotation. A strong theoretical justification is provided to demonstrate the effectiveness of the proposed TAILOR algorithm using regret analysis. The extensive evaluation conducted on various multiclass (imbalanced) and multilabel datasets demonstrates the ability of the proposed technique to produce comparable/better accuracy compared to the data-specific best candidate active learning strategy.


**Strengths:**

1. This paper has unified the strength of existing active learning algorithms into the single TAILOR framework. This avoids the burden of finding an effective active learning algorithm for a given specific problem.
2. The authors have provided sufficient novelty in this paper. Specifically, considering the deep active learning setting, the reward function design along with the bandit setting seems novel and non-trivial to me.
3. The extensive experimentation is conducted considering multiple datasets under  multiclass and multilabel settings. This enhances the credibility of the effectiveness of the proposed TAILOR framework.
4. The theoretical analysis showing better regret bounds for the proposed TAILOR algorithm compared to linear contextual bandits seems to be a very important contribution.

**Weaknesses:**

1. The experimentation is conducted using a single ResNet-18 architecture. Having experimental results on multiple architectures, especially more powerful ones, such as ResNet101, ViT would be helpful to even strengthen the effectiveness of the proposed technique.
2. The impact of \gamma (line 281)  is missing. I would expect to see the performance variation with respect to different \gamma values (some ablation study would be helpful).
3. One of the crucial components is the reward that is designed to maximize the participation of the data samples from the minority (rarest) classes in an imbalanced data setting. It would be interesting to see how the involvement of those minority (rarest) class data samples affects minority class performance. An ablation study comparing the rarest class performance of the proposed TAILOR framework with regard to different baselines would be interesting.

**Questions:**

1. In the rebuttal, it would be interesting to see experimental results mentioned in the weakness.

**Limitations:**

See Weaknesses.

---

> ### Author Rebuttal · Authors · 2023-08-09
>
> We would like to thank the reviewer for the detailed review and we would like to address the reviewer's concerns below.
>
> ### Ablation Study on Rarest Class Performance
>
> As shown in the PDF attached to our overall rebuttal (Figure 3), TAILOR significantly outperforms all other methods in improving the rarest class average precision. We will include this ablation study in the final version of our paper.
>
> ### Ablation Study on Discount Factor Gamma
>
> We agree with the reviewer that this would be a valuable experiment to run and we will include an ablation study in the final version of the paper.
>
> ### Additional Experiments on Larger Network Architectures
>
> Due to resource limitations, running active learning experiments that constantly retrain large neural network architectures can be prohibitive. We agree with the reviewer that these would add value to the paper, but cannot currently accommodate such experiments. As many existing papers in deep active learning use models only as large as ResNet18, we hope this meets the bar for acceptance. We will also add a note in future work and limitations section regarding model architectures.

---

> > ### Comment · Reviewer_NzMT · 2023-08-19
> > **Thanks for the Rebuttal**
> >
> > Thanks for the rebuttal, I would like to keep my current score. Also, I would like to see the ablation study result and large architecture result in the revised paper.

---

> > > ### Author Response · Authors · 2023-08-21
> > >
> > > We would like to thank the reviewer for the insightful comments which has been very helpful in improving our paper.

---

### Official Review · Reviewer_Fgsj · 2023-07-03

**Soundness:** 3 good
**Presentation:** 3 good
**Contribution:** 2 fair
**Rating:** 6
**Confidence:** 3

**Summary:**

This paper proposes a new method for algorithm selection in active learning. The problem is treated as a multi-armed bandit problem and the proposed method is based on Thompson sampling. Extensive experiments are conducted on several datasets, including both multi-class and multi-label classification setting.

**Strengths:**

1. The paper is well-written and easy to follow.
2. The new reward functions designed for label balance are interesting.
3. The authors provide regret analysis and the proposed method enjoys better regret bound than the standard linear contextual bandit algorithm.
4. The experimental results are comprehensive, including several benchmarks and two settings. TAILOR is comparable to the best candidate algorithm and even performs better than the best candidate for multi-label setting.

**Weaknesses:**

1. My major concern is the motivation behind active learning algorithm selection. For all the multi-class datasets, BADGE is almost always the best algorithm. Are there any datasets where BADGE does not perform well?
2. What's the time efficiency of TAILOR? Is TAILOR less efficient than the single algorithm?

Minor:
The figure is somewhat difficult to read the specific numbers. It would be better to provide an additional table.

**Questions:**

1. Are there any datasets where BADGE does not perform well?
2. Can BADGE be adapted for multi-label setting?
3. What's the time efficiency of TAILOR? Is TAILOR less efficient than the single algorithm?

---

> ### Author Rebuttal · Authors · 2023-08-09
>
> We would like to thank the reviewer for the detailed review and we would like to address the reviewer's concerns below.
>
> ### Motivation Behind Algorithm Selection (Why not always BADGE?)
>
> There are many cases BADGE is not the best. For example, under class imbalance, BADGE has been shown to underperform the best algorithms as shown in papers such as "SIMILAR: Submodular Information Measures Based Active Learning In Realistic Scenarios" by Kothawade et al. and "GALAXY: Graph-based Active Learning at the Extreme" by Zhang et al. on almost all datasets. The reasoning behind the success of such methods is primarily due to the importance of collecting a class-diverse set of labels. On the other hand, BADGE promotes diversity in the representation/gradient space, which does not guarantee a class-diverse labeled set.
>
> Moreover, in a recent paper that just came out less than two months ago, "LabelBench: A Comprehensive Framework for Benchmarking Label-Efficient Learning", the authors found that BADGE sometimes even underperform margin sampling in balanced settings when one fine-tunes from large pretrained models. In the imbalanced dataset iWildCam, BADGE is also shown to perform poorly in this paper.
>
> Overall, we believe the algorithm selection problem is a very important problem even beyond imbalance settings. This problem also arises from our experience deploying active learning on new applications, where it is very unpredictable which algorithm will even work better than random sampling.
>
>
> ### Adapting BADGE to Multi-label
>
> We think this is a very interesting problem, which deserves a paper on its own. However, as we have mentioned above, BADGE does not tend to work well in imbalanced settings, so we do not feel the necessity to adopt such an algorithm to multi-label classification in our setting.
>
>
> ### Time Efficiency of TAILOR
>
> We refer the reviewer to Appendix C for a detailed discussion of the time complexity of TAILOR. As a practical note in all of our experiments, TAILOR is more than 20% faster than the slowest candidate algorithm as it selects a diverse set of candidate algorithms instead of running a single algorithm the entire time. We would also like to note that the most significant time complexity often lies in neural network retraining. The retraining time dominates the running time of all algorithms including TAILOR.

---

> > ### Comment · Reviewer_Fgsj · 2023-08-19
> >
> > Thanks for your response. I have no further questions and I have raised my score to 6.

---

> > > ### Author Response · Authors · 2023-08-21
> > >
> > > We are glad that we have been able to address all of the reviewer concerns. We thank you for the insightful comments which has been very helpful in improving our paper.

---

### Official Review · Reviewer_LjXi · 2023-07-03

**Soundness:** 3 good
**Presentation:** 3 good
**Contribution:** 3 good
**Rating:** 6
**Confidence:** 2

**Summary:**

The paper proposed TAILOR, a Thompson Sampling framework for active learning algorithm selection for unlabeled, possibly imbalanced datasets by framing it as a multi-arm bandit problem. The authors compared with random Meta and another meta-framework ALBL, along with other AL methods on 10 multi-class multi-label vision datasets and concluded their framework new art.

**Strengths:**

Originality:

The work is deemed original based on its discussion with related work.

Clarity & quality:

The work is well written with great graphics like Figure 1 to illustrate key ideas. The appendix is also very helpful information.

Significance:

The work is relevant to the broader community of active learning and computer vision. The technical contribution it makes is solid in terms that the TAILOR does show improved AL efficacy compared with individual AL methods.

**Weaknesses:**

I have a suggestion for clarity: Add / explain what is the pool of AL algorithms that TAILOR considers at the very beginning of the paper (perhaps the intro part) as the plot shows 81 algs are considered (which can raise confusion for readers)

My major concern about the paper lies in the necessity of the different pieces of the reward function. I think an ablation study on the three different types of rewards function would do great help towards readers' understanding of relative contributions of their modules.


**Questions:**

1. As an active learning practitioner, my experience with active learning strongly signifies a non-stationarity in the effectiveness of different AL methods at different label levels (they are also path dependent, not only the number of labels but which labels were annotated till this iteration matters for the next round reward). Therefore I would like to ask how the authors bake this non-stationarity into the current formation.

2. What exactly are the 40 algs in the uncertainty (40 algs)? I didn't find it in Appendix F.




**Limitations:**

No potential negative social impact of this work is observed. The authors are encouraged to address their limitations more.

---

> ### Author Rebuttal · Authors · 2023-08-09
>
> We would like to thank the reviewer for the detailed review and we would like to address the reviewer's concerns below.
>
> ### Clarification on Reward Design and Choices:
>
> Regarding the choice of reward function, we believe this largely depends on the goal of using AL in practice. If the practitioner would like to maximize accuracy, using the class-diverse reward is the natural choice. If the practitioner would like to solve an active search problem for multi-label classification, then one would use the corresponding active search reward. Additionally, our framework and theoretical analysis allows the flexibility of choosing reward functions beyond the two reward functions mentioned above. As an example, in certain medical applications, certain classes may be much more important. By designing the proper weighting vector, our algorithm can be adapted to such applications. We will add part of the above clarifications as part of section 4.1.
>
> For the ablation study, we have included a comparison between class diverse reward and active search reward. As shown in figures 1&2 in the PDF attached in our overall rebuttal, the class diverse reward generates better mAP whereas the active search reward promotes the algorithms to annotate as many labels as possible. This trend is consistent in all of the datasets we have tested on. Also, note that for the class diverse reward, since we are interested in improving accuracy, we do not include the active search algorithms as candidate algorithms. The active search is a fundamentally different problem than active learning, since one almost want to annotate examples that are most certain to be in rare classes.
>
> ### Clarification on Candidate Algorithms:
>
> We agree with the reviewer that we should include more discussion on how candidate algorithms are constructed in the main text. The current form was primarily due to the page limit, and we will certainly take advantage of the extra page for the final version. We will also include part of the below discussion in the main text as the reviewer points out.
>
> To clarify on why the uncertainty algorithms have 40 individual algorithms, we construct $K$ uncertainty sampling algorithms for each dataset. Here, $K$ is the number of classes, and the $i$-th uncertainty sampling algorithm chooses the most uncertain examples for the $i$-th class. The reviewer can think of it as decomposition into $K$ individual binary classification problems. When running uncertainty sampling baseline, we run each algorithm approximately equal number of times. This design allows each class to be sample approximately equally. We would also like to note that the baseline EMAL computes the average of the uncertainty scores across all classes and samples the examples that are overall most uncertain across classes.
>
>
> ### Discussion on non-stationarity:
>
> Our experience highly aligns with what the reviewer is describing. In our paper, the non-stationarity comes to play in two regards. First, the weighting vectors $v$ changes from batch to batch. Next, since some algorithms may collect rare examples early on while others start collect rare examples later in the process, we added the discount factor gamma as described in "Implementation Details." of section 6.1. Since it is very hard to directly and effectively measure the generalization performance gain in the practical high batch size, low number of batch settings, we believe the class balancedness is a good proxy. Our above methods for handling non-stationarity seems sufficient according to our experiments.

---

> > ### Comment · Reviewer_LjXi · 2023-08-20
> >
> > Thank the authors for their response! I think the clarification on the candidate algorithm part is especially important and I highly recommend adding that part to the appendix or revise the main text using the extra page provided. For the quality of the work, I think it aligns with my original score and would like to keep my score.

---

> > > ### Author Response · Authors · 2023-08-21
> > >
> > > We will definitely clarify the candidate algorithms further in the final version. Thank you for the insightful comments which has been very helpful in improving our paper.

---

### Official Review · Reviewer_4Aq9 · 2023-07-11

**Soundness:** 2 fair
**Presentation:** 3 good
**Contribution:** 2 fair
**Rating:** 5
**Confidence:** 4

**Summary:**

Selecting the most appropriate active learning algorithm for a given dataset poses a significant challenge when applying active learning in real-world scenarios. This challenge stems from the fact that the performance of different active learning strategies could vary significantly across various scenarios and datasets. The paper introduces an interactive and adaptive algorithm selection strategy inspired by the concept of multi-armed bandit. By employing carefully designed reward functions, the proposed strategy can identify a subset of acquisition functions that yield maximum rewards. The authors prove that their selection algorithm exhibits a tighter bound on the Bayesian regret. Experimental evaluations conducted on both multi-class and multi-label classification tasks illustrate the promising performance of the proposed approach in terms of accuracy.

**Strengths:**

* Given the diverse nature of datasets, it is crucial to carefully consider and evaluate various active learning algorithms to achieve optimal performance and effectiveness. The concept of reducing the selection of acquisition functions to a multi-armed bandit problem is interesting as it allows the election process to incorporate accumulated rewards, providing a valuable approach for algorithm selection.
* The proposed method demonstrates adaptability to both multi-class and multi-label classification tasks by utilizing well-designed reward functions that suitably capture the task-specific requirements.
* Additionally, the authors present a proof establishing that the proposed selection algorithm outperforms an existing algorithm with a higher bound on Bayesian regret, although the reviewer did not thoroughly examine the soundness of the proof.

**Weaknesses:**

* The class imbalance issue is dealt with through a proper designed reward function, discussed in section 4.1. The idea of the inversely weighting each class based on its samples is not supervising, given that the weight scheme has been widely used in imbalanced semi supervised learning. The main concern is rather the empirical studies. Even though Figure 3(c) seems to show that the proposed method could deal with the imbalance problem, there is lack of studies about how tolerate the proposed method to different class imbalance ratios. Instead of showing the number of samples in the rarest class, it is better to plot the imbalance ratio. Furthermore, it is not clear how the imbalance issue is handled in the multi-label classification tasks.
* With respect to Assumption 3.1, it basically assumes to choose the top B samples ranked based on, for example uncertainty/least confidence scores. However, there are other methods, in particularly, some diversity-based AL methods, that choose the samples using clustering, which cannot be directly converted into an iterative selection process.
* The efficiency aspect of the proposed method is not adequately discussed in the paper. Consideration of efficiency factors, such as computational cost or time complexity, would provide a more holistic evaluation of the proposed method's practicality and real-world applicability.


**Questions:**

* For each iteration of acquisition, B samples are selected from B acquisition strategies, it sounds like an ensemble approach, doesn’t it? Different acquisition functions might capture, for example, different types of uncertainties, some to do with the model parameters, some to do directly with the model performance. Will this contribute to the performance difference?
* The reward function focuses on the class diversity in order to deal with the class imbalanced issue. It would be interesting to design some reward function that is closely related to the model performance.
* It is good to show the higher bound. However, can the author prove the convergence of the classifier, as done in WMOCU (Guang Zhao, Edward Dougherty, Byung-Jun Yoon, Francis Alexander, and Xiaoning Qian. Uncertaintyaware active learning for optimal Bayesian classifier. In International Conference on Learning Representations, ICLR 2021, 2021)
* In figure 3, there is no substantial difference between TAILOR and some other methods, including BADGE, can the author explain this? Meanwhile, BADGE is a diversity-based method using clustering, how this can be adopted in TAILOR with Assumption 3.1?

**Limitations:**

The author briefly touches on the limitations in section 6.3 and section 7. As mentioned above, it is also good to discuss more about the computational cost which could potential benefit the AL practitioners.

---

> ### Author Rebuttal · Authors · 2023-08-09
>
> We would like to thank the reviewer for the detailed review and we would like to address the reviewer's concerns below.
>
> ### Clarification on Diversity based algorithms
> As we have noted in Appendix A.1, all of the popular diversity-based algorithms rely on iterative procedure in choosing examples one at a time. For BADGE, the K-Means++ algorithm is an iterative procedure that produces cluster centers one at a time. This is very different from the K-means algorithm where the centers are produced all at once and dependent on the total number of clusters. Similarly, the greedy K-center algorithm for CoreSet and greedy optimization of SIMILAR are all iterative procedures that will choose the examples sequentially independent of the budget. We also point the reviewer to the implementation of BADGE in our repo as well as the original BADGE repo, which share the same core pieces of code. In short, while all such diversity algorithms tells a story of clustering, they all use greedy algorithms that are sequential and iterative in their implementations.
>
>
> ### Results on Imbalance Ratio
> We would like to thank the reviewer for suggesting this metric. We think this is indeed a valuable metric. As shown in the PDF attached in our overall Rebuttal, the imbalanced ratios follows a similar trend as the size of the rarest class. We only include 6 datasets due to page limit and will include all datasets in the final version. We also see a clearer gap by this metric. We also would like to point the reviewer to Table 1 in our paper, which details different imbalance ratios ranging from .5 to .01 for all of the datasets we experiment on in this paper.
>
>
> ### Clarification on Time Efficiency
> We refer the reviewer to Appendix C for a detailed discussion of the time complexity of TAILOR. As a practical note in all of our experiments, TAILOR is more than 20% faster than the slowest candidate algorithm as it selects a diverse set of candidate algorithms instead of running a single algorithm the entire time. We would also like to note that the most significant time complexity often lies in neural network retraining. The retraining time dominates the running time of all algorithms including TAILOR.
>
>
> ### Reward Design
> Having a reward that directly measures performance improvement is likely impossible for the practical scenarios we are considering, where examples are annotated in large batches with very few batches in total. As mentioned in our paper, this is a scenario we as practitioners face every day due to high retraining cost and setups of annotation infrastructures. This is also a scenario studied in the seminal paper "Batch Active Learning at Scale". Performance signals are sparse -- they are measured once for each batch. On the other hand, class-balancedness signals are per-example. Due to the limited number of batches in total and large number of candidate algorithms, a reward signal that is measured per-batch is insufficient to distinguish which algorithm is better. As a result, the best proxy reward we could think of was class-balancedness. In our experiments and prior works such as SIMILAR and GALAXY, this seems to consistently and strongly correlate with better performance under imbalance settings. In general, we totally agree with the reviewer that having a reward that directly measures performance would be great, but due to the limitations discussed above, we were unable to come up with any meaningful/effective reward design. We'd be happy to hear any concrete reward designs the reviewer has in mind.
>
> ### Proof of convergence
> The paper suggested by the reviewer proposes a specific active learning algorithm that has convergence guarantees. However, we study the very different problem of algorithm selection where we adaptively choose from a set of candidate active learning algorithms. Since the set of candidate algorithms (uncertainty sampling, BADGE, GALAXY, etc) do not have such convergence guarantees, we doubt the same type of convergence can translate to our setting. Our upper bound is on the algorithm selection aspect of the problem. Moreover, due to the non-stationarity of what each candidate algorithm choose to label (an active learning algorithm may choose examples differently early on versus later, see Reviewer LjXi's review on their practical experience also), we doubt "convergence" is even the right goal to pursue.
>
> We also want to note that the paper by Zhao et al. is largely a theoretical paper where they do not even experiment on deep neural networks. On the other hand, our paper has strong empirical experiments and the candidate algorithms we choose are deep active learning algorithms proven to be empirically effective.

---

> > ### Comment · Reviewer_4Aq9 · 2023-08-19
> >
> > Thanks very much for the authors' response.  The K-mean algorithm is often used in diversity-based active learning strategies to enhance the diversity of the acquired batch of samples in the bached setting. The reviewer still has a concern about data imbalance, particularly how the proposed algorithm can deal with data imbalance in the multilabel setting. meanwhile, Is it realistic to acquire 100,000 in real-world settings? Further, there is no substantial performance difference between BADGE and the proposed method, as shown in Figure 3, which is not answered in the responses. It is also good to include BADGE in the studies of data imbalance and rare classes. Overall, it is an interesting paper, the reviewer thus would like to keep the score.

---

> > > ### Author Response · Authors · 2023-08-19
> > >
> > > We thank the reviewer for the further clarification. We really appreciate the reviewer for pointing out these points and help us improve our paper.
> > >
> > > We agree if K-means is used in practice, then our framework cannot accommodate such setting. We will include a discussion in our final version.
> > >
> > > Regarding budget size, our real world applications constantly annotate millions of examples every month. We believe these are cases active learning really shines as saving even 10% of the annotation cost is a significant gain.
> > >
> > > ### Motivation Behind Algorithm Selection (Why not always BADGE?)
> > > For comparison against BADGE, we apologize we missed this comment. We have also included BADGE in all of our multi-class studies of data imbalance and rare classes. Could the reviewer clarify where we could further include BADGE? Regarding BADGE performing among the top, Reviewer Fgsj had the same concern that we have addressed. We attach the same reasoning below:
> > >
> > > First of all, in multi-label classification, BADGE is no longer a valid algorithm and we have shown the utility of TAILOR in such settings.
> > >
> > > In multi-class settings, there are many cases BADGE is not the best as shown in existing literatures. For example, under class imbalance, BADGE has been shown to underperform the best algorithms as shown in papers such as "SIMILAR: Submodular Information Measures Based Active Learning In Realistic Scenarios" by Kothawade et al. and "GALAXY: Graph-based Active Learning at the Extreme" by Zhang et al. on almost all datasets. The reasoning behind the success of such methods is primarily due to the importance of collecting a class-diverse set of labels. On the other hand, BADGE promotes diversity in the representation/gradient space, which does not guarantee a class-diverse labeled set.
> > >
> > > Moreover, in a recent paper that just came out less than two months ago, "LabelBench: A Comprehensive Framework for Benchmarking Label-Efficient Learning", the authors found that BADGE sometimes even underperform margin sampling in balanced settings when one fine-tunes from large pretrained models. In the imbalanced dataset iWildCam, BADGE is also shown to perform poorly in this paper.
> > >
> > > Overall, we believe the algorithm selection problem is a very important problem even beyond imbalance settings. This problem also arises from our experience deploying active learning on new applications, where it is very unpredictable which algorithm will even work better than random sampling.
> > >
> > > ### Imbalance in Multi-Label Setting
> > > We deal with imbalance in multi-label setting by up-sampling the classes that appear the rarest. Specifically, the number of examples labeled with class-i simply counts all the examples with class-i as one of its labels. The inverse weighting is then based on this count, which encourages examples with rare class label(s) to be upsampled. We hope the reviewer can further clarify their confusion, so we can address such concerns in our paper.
> > >
> > > Thank you again for the insightful discussion.

---

> > > > ### Author Response · Authors · 2023-08-21
> > > >
> > > > We hope our latest comment have cleared up some confusion from the reviewer. Regardless, we would like to take the time to thank the reviewer for the insightful comments and engagement, which has been very helpful in improving our paper.

---

### Official Review · Reviewer_Ra5K · 2023-07-25

**Soundness:** 2 fair
**Presentation:** 1 poor
**Contribution:** 2 fair
**Rating:** 3
**Confidence:** 3

**Summary:**

This paper proposes an algorithm, TAILOR, that iteratively and adaptively selects candidate active learning algorithm to gather class-balanced examples. Experimental results demonstrate that TAILOR achieves comparable or better performance than the best of the candidate algorithms.

**Strengths:**

The paper presents a new active learning algorithm and provides some theoretical analysis. Good empirical results are reported.

**Weaknesses:**

The presentation of the paper leaves much to be desired, and its contribution appears to be quite limited. Additionally, the paper contains improper or insufficiently supported claims, such as “meta algorithm”, “focus on class imbalanced setting”, and “the first adaptive algorithm selection strategy”.


1. The organization and content selection are unsatisfactory.

(1) The introduction is not well motivated or presented. The framework illustrated in Figure 1 is an adaptive active learning procedure, and it is difficult to build connection with a multi-armed bandit problem from such a figure. It is not very proper to report a result figure in the introduction as well.

(2) Algorithm 1 is not the contribution of this paper but was included on page 4 with a lot of space.  Since most contents of section 3.2 are from the literature, the presentation can be largely compressed.

(3) Much space of the paper is not on the main contribution of the paper. The proposed algorithm TAILOR is only presented in 4.3.

(4) Throughout the overall paper (in particular section 4), method presentation, previous work review, and contribution claims are jumbled together without a clear and logical structure of presentation.



2. The presentation of the paper lacks sufficient clarity.

(1) In 4.2, the notations and concepts are not well explained. For example, what is the 1-sub-Gaussian distribution?  What does it mean by stating “Nature reveals weighting vector v^t”?

(2) Section 4.2 also fails to present a clear and principled connection of the proposed active learning setting to the bandit problem.

(3) The Equations are not well explained. For example, the “\lor” operator in the class diversity reward is not described.



3. Authors claim the algorithm as a meta algorithm. The TAILOR algorithm in Algorithm 2 didn’t show any meta-learning process. The meta concept needs to be clarified.



4. The paper claims on focusing on class imbalanced setting,  but  fails to provide any substantial discussion or analysis on this aspect, except for conducting experiments on datasets with class-imbalance.



5. The reward functions proposed in 4.1 are heuristic and lack principled connections to the model performance. There are no discussions on how to choose reward functions on different datasets.



6. The claim of “the first adaptive algorithm selection strategy for deep active learning” is not proper. Adaptive active learning has been studied in the active learning literature. Most active learning strategies developed in the literature are not dependent on the classifiers, and hence there is no need to distinguish linear models from deep models from the active learning perspective. In particular, the proposed active learning method has no special ties to deep models.



7. Many related works on adaptive active learning or meta active learning are missing, e.g., [1,2,3].
[1] “Adaptive Active Learning for Image Classification”. 2013.
[2] “Active Learning with Multi-label SVM Classification”. 2013.
[3] “Meta-Learning for Batch Mode Active Learning”, 2018.


**Questions:**

Please see the weaknesses above.

---

> ### Author Rebuttal · Authors · 2023-08-07
>
> We would like to thank the reviewer for writing a detailed review, but the reviewer appears to have some fundamental misunderstanding of what our paper is studying.
> ### Research Problem and Related Work
> We are not proposing another active learning algorithm. Our goal is to come up with an online and adaptive way to choose from hundreds of promising active learning algorithms during deployment.
>
> To this end, we are indeed the first to study this problem for deep neural networks. Regarding related work,
> 1. All active learning algorithms are by nature adaptive. We hope the reviewer can clarify what "adaptive active learning" means and how it differs from standard active learning papers.
> 2. The two "adaptive active learning" papers suggested by the reviewer are individual AL algorithms that could serve as candidate algorithms in our framework. Our paper studies algorithm selection instead of introducing another AL algorithm. While there are thousands of AL algorithms proposed, we chose a set of popular candidate algorithms in the deep AL era. The algorithms are tested to be effective in practice and considered as "standard".
> 3. We would also like to point the reviewer to our related work section where we discuss why meta algorithms proposed for algorithm selection in the past do not work in the era of deep learning. For our problem, there is a fundamental difference between linear models and deep neural nets. While many of the past work are not even computationally feasible for deep learning, we demonstrate in our experiments that "Active Learning by Learning" by Hsu et al. is no better than choosing candidate algorithms at random, and significantly underperforms TAILOR.
> 4. While we use the term "meta algorithm", it bears minimal relevance to "meta learning", a term only popularized in the past few years. "Meta algorithm" simply refers to an algorithm that controls the procedure to run a pool of algorithms, by its literal meaning. The "meta active learning" paper suggested by the reviewer studies a very different problem from ours. The relatedness is not immediate to us other than common wording in the title. We hope the reviewer can elaborate on the relatedness and we would be more than happy to include the "meta AL" literature if they are indeed related to the problem we are studying.
> ### Novelty and Contribution
> 1. Figure 1 is not a figure on adaptive active learning, but instead adaptive algorithm selection for active learning.
> 2. Algorithm 1 and section 3.2 is a novel summarization of the past literature of Baram et al, Hsu et al, and Pang et al. While the concepts introduced are not novel, the generality of the framework is. We also feel the importance to explain our setting of "algorithm selection" clear by presenting this section. Again, this is not the standard AL setting but rather a less studied yet very important problem. We would also like to note that many papers spend more than one or even two pages to describe the backgrounds and formulations of prior work, which we only took less than half a page in 3.2 (Assumption 3.1 is novel).
> 3. The rewards, bandit reduction, TAILOR algorithm, analysis and empirical experiments are all novel contributions. Again, we are neither studying the standard AL problem nor proposing another AL algorithm.
> ### Reward Design and Choices
> To further elaborate on connections between our reward design and accuracy performance, the goal of our meta algorithm is to adaptively select the algorithm(s) that produce a class-balanced dataset. In general, labeling a more class-balanced dataset translates to better generalization performance in balanced accuracy, as observed in prior work such as SIMILAR (Kothawade et al.) and GALAXY (Zhang et al.), which are cited in the paper. The theoretical analysis also upper bounds the cumulative regret, which is the summation over rewards designed to promote class-diversity in the labeled set. We empirically demonstrate the importance of having a class-balanced labeled set and we are glad the reviewer thinks our experiments are convincing.
>
> Regarding the choice of reward function, we believe this largely depends on the goal of using AL in practice. The Class-diverse reward is the natural choice if the practitioner would like to maximize accuracy. If the practitioner would like to solve an active search problem for multi-label classification, then active search reward should be used. In addition, our framework and analysis allows the flexibility of choosing rewards beyond the two reward functions mentioned above. As an example, in certain medical applications, certain classes may be much more important. By designing the proper weighting vector, our algorithm can be adapted to such applications. We will add part of the above clarifications as part of section 4.1.
>
> Lastly, as mentioned by Reviewer 4Aq9, inverse weighted sampling has been proven successful in many contexts before. We'd argue it is a pretty standard and justified technique due to its wide adoption.
> ### Our Writing
> 1. We believe there's not a single "right way" of communicating science. As an example, OpenAI's CLIP paper "Learning Transferable Visual Models From Natural Language Supervision" has result figures in introduction.
> 2. We would also like to point the reviewer to the other reviewers' reviews, where they unanimously rated the paper with "good" or "excellent" presentations. Two of them explicitly says "The work is well written with great graphics like Figure 1 to illustrate key ideas" and "The paper is well-written and easy to follow."
> 3. $a \lor b := max(a, b)$, we will add this in the final version, thanks!
> 4. For the definition of 1-sub-Gaussian distribution, we point the reviewer to textbooks like "High-Dimensional Probability: An Introduction with Applications in Data Science" by Roman Vershynin. "Nature reveals ..." equivalently means "The algorithm is given ...". Both terms are common concepts and languages in the bandit community.

---

> > ### Comment · Reviewer_Ra5K · 2023-08-18
> >
> > I have read the authors' response and would keep my rating score.

---

> > > ### Author Response · Authors · 2023-08-21
> > >
> > > We would like to thank the reviewer for providing a detailed review.

---

### Author Rebuttal · Authors · 2023-08-09

We would like to thank all of the reviewers for their insightful reviews. Attached PDF includes all of the figure plots mentioned in our rebuttals to each individual reviewer.

---

### Comment · Area_Chair_4r6Y · 2023-08-13
**Author-Reviewer Discussion**

Thanks to the authors for submitting a detailed rebuttal. Can the reviewers please read the author response and let them know if they have any further questions / need any additional clarifications?

Best regards,
 - Your AC.

---

> ### Comment · Area_Chair_4r6Y · 2023-08-19
>
> Thanks to the authors and reviewers for the ongoing discussion. Can the reviewers please take a look at the author rebuttals and let them know if they have any other questions / need further clarifications?
>
> Best regards,
>  - Your AC.

---

### Decision · Program_Chairs · 2023-09-21

**Decision:**

Accept (poster)

**Comment:**

This paper was reviewed by 5 experts in the field and received one Reject, one Borderline Accept, two Weak Accept and one Accept as the ratings. The reviewers agreed that the paper addressed an important problem, and alleviates the burden of finding an effective active learning algorithm for a specific problem. Extensive experimental studies have been conducted on multiple datasets under the multiclass and multilabel settings. Theoretical analysis has also been conducted, showing better regret bounds for the proposed TAILOR algorithm compared to linear contextual bandits.

The reviewers raised concerns about the time efficiency of the proposed algorithm, ablation studies on the different types of reward functions and on the rarest class performance, which were addressed by the authors in the rebuttal. Questions were also raised about the performance of the algorithm in the multi-label setup with class imbalance, adoption of diversity based algorithms like BADGE in the proposed framework, which were also convincingly addressed by the authors during the post-rebuttal discussion period.

The reviewers, in general, have a positive opinion about the paper and its contributions. The AC appreciates the usefulness of the proposed method in appropriately identifying an effective active learning algorithm for a given problem, together with the theoretical analysis presented. Even though Reviewer Ra5K shares a negative opinion about the paper, the concerns raised were mostly about the organization and writing style of the paper. The AC believes that these issues can be addressed in the final version and should not be considered a reason for rejecting the paper.

Based on the reviewers’ feedback, the decision is to recommend the paper for acceptance to NeurIPS 2023. The reviewers have expressed some valuable concerns, particularly about the choice of the candidate algorithms, ablation study on the discount factor \gamma, experiments on large model architectures and the organization and presentation of the paper. The authors are encouraged to address these in the final version of their paper. We congratulate the authors on the acceptance of their paper!